

# A method to estimate freezing rain climatology from ERA-Interim reanalysis over Europe

Matti Kämäräinen[1], Otto Hyvärinen[1], Kirsti Jylhä[1], Andrea Vajda[1], Simo Neiglick[1], Jaakko Nuottokari[1], and Hilppa Gregow[1]

[1]Finnish Meteorological Institute, Erik Palménin aukio 1, P.O.Box 503, FI-00101 Helsinki, FINLAND

*Correspondence to:* Matti Kämäräinen (matti.kamarainen@fmi.fi)

**Abstract.**

A method for estimating the occurrence of freezing rain (FZRA) in gridded atmospheric datasets was evaluated, calibrated against SYNOP weather station observations, and applied to the ERA-Interim reanalysis for climatological studies of the phenomenon. The algorithm, originally developed for detecting the precipitation type in numerical weather prediction at the Finnish Meteorological Institute, uses vertical profiles of relative humidity and temperature as input. Reanalysis data in 6-hourly time resolution was analyzed over Europe for the period 1979–2014. Mean annual and monthly numbers of FZRA events, as well as probabilities of duration and spatial extent of events, were then derived. The algorithm was able to reproduce accurately the observed, spatially averaged interannual variability of FZRA (correlation = 0.93) during the 36-year period, but at station level rather low validation and cross-validation statistics were achieved (mean correlation = 0.44). Coarse grid resolution of the reanalysis, and misclassifications to other freezing phenomena in SYNOP observations, such as ice pellets and freezing drizzle, contribute to the low validation results at station scale. Although the derived gridded climatology is preliminary, it may be useful, for example, in safety assessments of critical infrastructure.

## 1 Introduction

Freezing rain (FZRA) is liquid, supercooled precipitation which freezes on coming into contact with solid objects, forming a coating of ice (World Meteorological Organisation, 2010, 2011). It is a relatively rare but high-impact wintertime weather phenomenon, and in Europe it affects mainly central, eastern and northern parts of the continent. Although major events resulting in heavy ice accretion are not as common as lighter, short-lived cases, the direct damages they cause to critical infrastructure (transportation, communication and energy), and forestry are substantial. For example, the ice coating formed on trees and powerlines causes them to fail, leading to severe power outages, transportation disruption, delays in emergency responses, and severe economic losses (Call, 2010; Lambert and Hansen, 2011). Lighter freezing rain events are also harmful because of their indirect effects, the most important being the reduced friction on road surfaces that results in increased rates of accidents, injuries, and difficulties in transportation (Degelia et al., 2015).

During recent years some major events have been experienced across Europe. On 31 January – 3 February 2014 a prolonged, heavy FZRA and blizzard event hit the Alpine region, Hungary and the Balkan Peninsula. In Slovenia, during these four days,



40–300 mm precipitation fell over and resulted in 10 cm accumulation of ice (Markosek, 2015). Severe damage was caused to critical infrastructure, e.g. 30 km of power lines were completely destroyed and 174 km were inoperative, while railways and road traffic were heavily disrupted or closed for several days. Over 0.5 million hectares of forest were damaged, and the total cost resulted in around 400 million euros (Vajda, 2015). In Croatia, over 80% of the population were left without electricity (Editorial, 2014). Other severe cases include the Moscow FZRA event on 25–26 December 2010, where flights were cancelled at the Domodedovo airport and power supplies to trains, trams, and busses were destroyed; and the 13 December 2012 case, where the British Isles and France suffered from a mix of freezing rain and snow.

Despite the severe impacts of FZRA events, only a few publications concerning the European climatology of this phenomenon exist currently. Carrière et al. (2000) provided a climatology of freezing precipitation (including FZRA, freezing drizzle and ice pellets) based on SYNOP weather station reports from three winters. Bezrukova et al. (2006) presented climatological information of FZRA over the former USSR, thus including eastern Europe. Most of the other studies focus on the occurrence of in-cloud or near-surface icing (e.g., Bernstein et al., 2009; Bernstein and Le Bot, 2009; Le Bot, 2004; Le Bot and Lassegues, 2004). More comprehensive studies have been undertaken on ice storm and FZRA climatologies over North America, including studies for various regions (Cortinas, 2000; Changnon, 2003; Cortinas et al., 2004), impacts of ice storms on different sectors (Proulx and Greene, 2001), and changes in FZRA climatology (Cheng et al., 2007, 2011; Lambert and Hansen, 2011; Klima and Morgan, 2015) as well as various wintertime precipitation types (Stewart et al., 2015), and impacts and details of FZRA storms and related synoptics (Hosek et al., 2011; Call, 2010; Roberts et al., 2008).

The two formation mechanisms of FZRA are rather well known. In the majority of cases a near-surface freezing layer with an accompanying melting layer above makes the hydrometeors – formed above these layers – to be in liquid, supercooled phase when they hit the ground and freeze on contact with objects at the surface (World Meteorological Organisation, 2010). Other FZRA cases occur without the cold layer – melting layer structure, as a result of the warm rain process (Rauber et al., 2000; Carrière et al., 2000), where collision and coalescence of the small droplets ensure the liquid form. The latter mechanism is usually associated with drizzle or freezing drizzle, but in some cases it may lead to formation of FZRA.

Several approaches have been developed for identifying wintertime precipitation types (e.g., snow, ice pellets, freezing rain) in numerical weather prediction (NWP) models, most of them for North America and many of them reviewed by Cortinas et al. (2002). With varying complexity, all of them are based on the vertical temperature profile which is used to predict the state of the hydrometeors in the atmosphere and on the surface level in particular. Usually the vertical humidity profile is used as well. For example, Ramer (1993) developed an empirical method which explicitly resolves the melting and freezing of the descending hydrometeors. This method has been widely used in NWP and in related studies (e.g. Reeves et al., 2014), showing a good skill among the other precipitation typing algorithms. A slightly more indirect, and perhaps simpler approach was presented by Bourgouin (2000), who estimated the phase of precipitation based on areas between the temperature profile and the 0°C isotherm on a tephigram.

Three-dimensional gridded meteorological datasets at daily or sub-daily temporal resolution, such as output from numerical climate models or reanalysis models, are commonly used in climate studies to account for gaps in time series of weather station data, and to fill the sparsely covered areas, like seas and the above-surface atmosphere. However, a variety of issues complicate





their use for estimating precipitation types. One important uncertainty arises from the fact that it is not straightforward to compare point-like weather station observations (representing local climate) and grid cells (representing climate of a larger area). As shown e.g. by Stewart et al. (2015), Reeves et al. (2014) and Ryzhkov et al. (2014), even small details in the vertical distribution of temperature can affect the surface precipitation type. However, because a gridded dataset typically has a rather

coarse spatial resolution and a smoothed orography, its vertical temperature structure may differ locally from the reality. Similarly, very minor modelling uncertainty, or natural uncertainty related to subgrid-scale processes, might cause the predicted temperatures of the freezing or melting layers to be slightly off from the values that would lead to FZRA.

In this study we introduce a freezing rain detection algorithm, originally developed and operationally used in the numerical weather prediction at the Finnish Meteorological Institute ($FMI_{NWP}$) and here implemented for climatological applications.

The algorithm was applied to the ERA-Interim reanalysis data (Dee et al., 2011) to provide a climatology of FZRA in Europe for the period 1979–2014. First, the SYNOP weather station and reanalysis datasets used in calibration of the algorithm are introduced, along with an optimization-based calibration procedure; second, the calibration results are validated using multiple approaches; and lastly the climatology is produced and analysed shortly. In the analysis the following statistics are focused on: the total number of 6-hourly FZRA cases at each station or grid cell during the 36-year study period, and the average frequency

of these cases per location in the whole study domain and in three subregions as a function of time. In addition, duration and spatial extent of the FZRA events are studied.

## 2  Materials and methods

The $FMI_{NWP}$ algorithm uses threshold values of the air temperature and humidity in the near-surface freezing layer and in the above melting layer to distinguish the FZRA events from non-events. In order to estimate the FZRA climatology in Europe

based on reanalysis data, these threshold values needed to be reconsidered for two reasons. First, some of the original threshold values were subjectively selected by meteorologists in Finland, which involves uncertainty related to subjective decisions. Second, the values are likely to be somewhat sensitive to potential biases in temperature and humidity, and these biases may be different in NWP data and in reanalysis data. Consequently, a calibration procedure was developed that employed SYNOP weather station observations to redefine the threshold values of the parameters, as discussed below. The calibrated version of

the $FMI_{NWP}$ algorithm, used in the climatological analysis of FZRA, is denoted here by $FMI_{CLIM}$ algorithm.

### 2.1  SYNOP weather station data

For validation and algorithm-calibration purposes, the observed occurrence of FZRA events was derived from 3-hourly SYNOP weather station recordings. Data from 4000 manually operated stations were collected from the Meteorological Archival and Retrieval System (MARS) of the European Centre for Medium-Range Weather Forecasts (ECMWF). Automated stations were

not accepted due to their reduced ability to distinguish different types of precipitation (Sheppard and Joe, 2000). The present weather part of a SYNOP observation (World Meteorological Organisation, 2011) consists of one hundred codes describing the most important weather at the time of observation and one hour before it. In this study the WMO codes directly referring





to FZRA were selected to represent the phenomenon: 24 (freezing rain within past hour but not at observation time), 66 (light freezing rain) and 67 (moderate to heavy freezing rain).

To be included in this study, the stations were required to contain a valid present weather code in >80% of the 3-hourly time steps during the period 1979–2014. If stations with shorter or less regular records had not been excluded, the observed

number of FZRA events per station might have been distorted. Besides, regularly working and maintained stations with high-frequency observations are assumed to be more reliable. Altogether 525 stations out of 4000, presented in Fig. 1, passed these first conditions.

For calibration and validation purposes, and for further analyses of FZRA in station locations, the data was filtered further:

- In many countries of eastern Europe, the FZRA has been interpreted differently compared to other European coun-
tries, so that FZRA is reported only in the presence of simultaneous glaze ice (Bezrukova et al., 2006). This leads to underestimation of the observed number of cases, and for this reason most of these stations were excluded.

- The stations located above 2000 m above sea level and those having less than 10 FZRA observations were excluded, as the algorithm does not have enough pressure levels for high elevations, and reliable observations might be difficult for observers with limited experience of the phenomenon due to its rarity.

- To exclude grossly erroneous recordings, the FZRA observations with the surface temperatures below -30°C or above +10°C were rejected.

Applying the above mentioned restrictions excluded 325 stations (circles in Fig. 1) and thus reduced the total number of stations to 200. These remaining stations, located predominantly in northern and central Europe, are used in the calibration and validation process of this study. Finally, the time steps 00Z, 06Z, 12Z and 18Z were picked from the 3-hourly SYNOP

observations to allow direct comparisons of the present weather codes with predictions of the FZRA events that were derived with the $FMI_{CLIM}$ algorithm using the 6-hourly ERA-Interim reanalysis data. The comparisons were conducted both for the individual stations and for spatially clustered stations.

In order to divide the stations into clusters, variables which best explain the spatial differences in the total number of observed FZRA cases in 1979-2014 were sought. Therefore, different linear models were fitted. Strictly speaking, the number of cases

are count data (non-negative integer values from counting) and should be modelled using the nonlinear Poisson regression, but this did not change our conclusions and linear models are somewhat easier to understand. Finally, the best variables were used to classify stations into subgroups, and further analyses were performed for these groups separately. The variables studied were the distance to the nearest coastline (NASA, 2009), station elevation, and longitude. Modelling the number of cases with only one variable, the distance from the coastline was the best in explaining the variance (the adjusted $R^2 = 0.13$, $p << .001$).

Adding another variable or interaction term did not improve the results substantially, while a simple stepwise model selection using Akaike information criterion (Venables and Ripley, 2002) selected also the station elevation, the adjusted $R^2$ did not change. To keep the analysis simple, only the distance to the nearest coastline was thus used for the classification. The stations were then grouped to "coastal" (0–140 km), "semi-coastal" (140–330 km), and "continental" (>330 km). Boundaries of classes





were selected simply so that each group contained equal number (67) of validation stations. Boundaries of classes are shown in Fig. 1.

## 2.2 ERA-Interim reanalysis data

Relative humidity and temperature from 925, 850, 700 hPa and 2-meter levels, surface pressure, and precipitation of the

ERA-Interim reanalysis dataset (Dee et al., 2011) were used as a predictor data. First, the data was bilinearly interpolated to station locations for calibrating and validating the FZRA detection algorithm, and after calibration, the climatology of FZRA was derived using the gridded data in the original $0.7° \times 0.7°$ resolution. Only the 6-hourly analysis part of the data was used except for the precipitation, where the 6-hourly forecasted part was used. The 12-hourly precipitation sums were transformed into 6-hourly sums.

When the predicted and observed FZRA events were compared at station level, time steps without SYNOP observations of present weather codes were excluded also from the interpolated reanalysis data, which ensured the comparability. Even though excluding those time steps leads to an underestimation of the total number of FZRA cases, the effect is presumably minor, as the stations were selected so that their time series were required to be at least 80% complete.

## 2.3 FMI$_{NWP}$ algorithm

The FZRA identification part of the precipitation typing algorithm (FMI$_{NWP}$), used at the weather service of the Finnish Meteorological Institute for numerical weather predictions, was adopted in this study. The algorithm uses temperature and relative humidity from four pressure levels (surface, 925, 850 and 700 hPa), and surface air pressure. Surface pressure is used to avoid analysing below-surface data in mountainous regions. The FMI$_{NWP}$ algorithm is originally used to predict locations where FZRA is conceivable, and as such it does not take the modelled precipitation intensity ($Pr$) into account. In our analyses

the precipitation intensity was included to identify the actual FZRA cases. For FMI$_{NWP}$ the $Pr$ value $0.04 \mathrm{\ mm\ 6h^{-1}}$ was selected so that the identified total number of FZRA cases, calculated as a sum over the whole study domain and all the years, corresponds the observed total of 7900 cases in the 200 stations.

A pseudocode representation of the algorithm is shown in Fig. 2. First, the preconditions for FZRA are checked: (1) the near-surface air temperature T2m $= T_{cold}$ has to be lower than its predefined threshold value $T_{cold}^{thr}$, and (2) the maximum temperature

of the above-surface layers $T_{max}$ and (3) the surface precipitation rate $Pr$ need to be higher than their threshold values ($T_{melt}^{thr}$ and $Pr^{thr}$, respectively). In the next step the upper level of the near-surface cold layer $p_{cold}$ is defined by selecting the pressure level closest to the ground surface, while additionally taking into account the minimum acceptable cold layer depth $h_{cold}^{thr}$ (in hPa) which ensures that the falling raindrops are properly supercooled. Finally the existence of a moist and warm melting layer is checked by investigating the layers above the cold layer. If temperature and humidity $RH$ in at least one of those layers are

above their predefined thresholds $T_{melt}^{thr}$ and $RH_{melt}^{thr}$, FZRA is predicted.

Compared to other precipitation detection algorithms, such as those presented by Ramer (1993) and Bourgouin (2000), the FMI$_{NWP}$ is presumably faster to implement and run, which makes it ideal for analysing large climatological datasets. Of these two algorithms, FMI$_{NWP}$ resembles Bourgouin (2000) more, mainly because the depth and temperature of the near-surface





cold layer ($h_{cold} = p_{surf} - p_{cold}$ and $T_{cold}$, respectively) together describe the energy required to supercool the raindrops. $\text{FMI}_{NWP}$ assumes that melting layer and the layer where precipitation is generated are the same, even though in reality they can be separated so that precipitation is formed above the melting layer.

### 2.4 Calibration

In calibration, each of the threshold parameters ($h_{cold}^{thr}$, $T_{cold}^{thr}$, $T_{melt}^{thr}$, $RH_{melt}^{thr}$ and $Pr^{thr}$) was discretized to cover the practical, realistic range, and the calibration was then performed in a multiple loop, where each combination of the parameters was tested using the algorithm and a suitable reward function.

  The comparison between the algorithm results and FZRA observations can be presented in a $2 \times 2$ contingency table (Table A1 in Appendix A), so standard verification measures (e.g., Jolliffe and Stephenson, 2012) can be used. Four candidates

were tested to find the appropriate reward function for optimization, namely the Proportion Correct (PC), the Critical Success Index (CSI), the Heidke Skill Score (HSS), and the Symmetric Extremal Dependence Index (SEDI; see Appendix A for the definition of these and other verification measures used in this study). All candidates are positively oriented: the higher the value, the better the agreement between the predictions and observations.

  When only one measure is used, the CSI is the best one, because strongly biased solutions automatically get worse CSI

values. However, all tested reward functions tend to favor biased solutions, either by overestimating (PC, HSS, SEDI) or underestimating (CSI) the total number of FZRA events over all stations and years. This is not desirable as the main interest of the study is in the occurrence climatology of FZRA. Therefore an additional, bias-dependent term was added, and the final form of the reward function was:

$$J = \text{CSI} - |\log \text{B}|, \tag{1}$$

where B is the bias (A7). The logarithm scales B from $-\infty$ to $+\infty$, the best, unbiased value being zero. Therefore biased solutions result a non-zero bias term that is then subtracted from CSI. Adjusted values used in the algorithm are presented in Table 1.

### 2.5 Climatological analysis

Climatological analysis was performed separately for SYNOP observations, ERA-Interim in station locations, and for ERA-

Interim in the original grid. The following statistics were calculated:

- Total numbers, mean annual numbers, and mean monthly numbers of 6-hourly FZRA events per station or grid cell in 1979–2014. In this analysis one FZRA event occurs when one station or grid cell encounters freezing rain in one time step.

- Spatially averaged annual mean numbers of FZRA events in 1979–2014. Spatial averaging was performed over stations

in subgroups, and over all stations. Definition of an event is the same as above.



- Duration of FZRA events separately for station data and for gridded data. In this analysis one FZRA event occurs when one station or grid cell encounters freezing rain in one or in successive time steps. Durations were calculated from a 1-dimensional vector containing the time series of stations or grid cells in a row. Station data with 200 stations and 53 000 6-hourly time steps comprises a total of 28 million data points. The gridded reanalysis data consists of 4000 grid cells and 53 000 time steps with altogether 210 million data points.

- Spatial extent of 6-hourly FZRA events separately for station data and for gridded data. In this analysis one FZRA event occurs when one or multiple stations or grid cells encounter freezing rain simultaneously. The spatial extent was calculated based on the number of impacted stations or, for gridded data, based on the spatial coverage of impacted grid cells over the domain, using an approximative 6400 $\mathrm{km}^2$ grid cell size.

After derivation of durations and spatial extents of events, empirical probability distributions (containing events and non-events) were formed and plotted against corresponding sorted values of the variables.

## 3 Results

In the following, we first have a look at the refined threshold values in the $\mathrm{FMI}_{CLIM}$ algorithm. The performance of the algorithm in predicting FZRA cases is then assessed using the observed weather station data, and finally, the findings concerning climatological features of FZRA are presented. The cross-validation results are based on calibrations in sub-periods, while other validation results are based on the final calibration.

### 3.1 Cross-validation of calibration

To study the sensitivity of the threshold values to the selection of the calibration period, a cross-validation framework was applied, where the total of 36 years of data was divided into five non-overlapping sub-periods. The calibration was then performed for each of them separately, using Eq. (1). Each sub-period contained a 29-year calibration part and a 7-year validation part so that the validation years were different in all sub-periods. The means calculated over the results from different calibration periods were used for the climatological analysis of FZRA: they are close to the values that were achieved using the whole 1979–2014 period for calibration.

The calibration changed most of the threshold parameter values only slightly (Table 1), which confirms that (1) the $\mathrm{FMI}_{NWP}$ algorithm performs as designed, and (2) no strong biases in mean values exist in the ERA-Interim data. For example, if the mean temperatures in some of the layers studied were far from the reality, the optimal value would have drifted away from the physically motivated $0\,^\circ\mathrm{C}$ limit in the calibration. As an exception, the minimum depth of the near-surface cold layer $h_{cold}^{thr}$ was notably altered by the calibration, as the optimal value appeared to be 65 hPa, which is over 300% larger compared to the original value, 15 hPa. In the lower troposphere these correspond roughly to the depths of 400 and 130 metres respectively. The bias-dependent part of the reward function (Eq. 1) was useful in stabilizing the calibration and excluding the less credible





combinations of threshold values. Without it, calibration introduced new biases (not shown) either in the annual number of the FZRA cases in subgroups (Fig. 3), or in the total number of cases per station (Fig. 1).

As Table 2 shows, the calibration enhanced most validation metrics, except bias and F. On average, the hit number $a$ was improved by 20%, and the false alarm $b$ and miss $c$ numbers both were reduced. While H clearly improved, the variability of
$b$ was rather large, and the change of F is not statistically significantly. CSI was improved by 20%, HSS by 20% and SEDI by 5% in calibration, and these changes were statistically significant. The absolute values of CSI and HSS are rather modest, but it is well known (Jolliffe and Stephenson, 2012) that these measures tend to zero when the base rate is very low, as is the case of FZRA. SEDI is designed for the evaluation of rare events and gives much higher values. Still, only one event in five is correctly detected (H $\approx$ 0.20).

## 3.2   Performance of the freezing rain detection algorithm

### 3.2.1   SYNOP weather code classification

The algorithm-based ($\text{FMI}_{CLIM}$) classifications of weather situations to FZRA events were compared with SYNOP observations of present weather at the validation stations. In Fig. 5a, the distribution of observed SYNOP codes, when an event was classified as FZRA, is presented. The distribution shows that only 22% of classified events coincide with SYNOP codes for
FZRA (codes 24, 66, and 67). About 30% of classifications coincide with codes when no rain of any kind was observed at the SYNOP stations (codes 2 and 10), although reanalysis data did imply rain of at least 0.3 mm/6h (Table 1). Codes associated with light precipitation of other types (drizzle, rain and snow, codes 56, 61, and 71) coincide with FZRA classification about 6% each. However, most of these codes occur much more often than codes associated with FZRA. In order to illustrate how the relative numbers of false alarms deviate between different SYNOP present weather codes, the proportion of classified FZRA
events in each code is shown in Fig. 5b. For SYNOP codes associated with FZRA, this proportion can be interpreted as the hit rate H (A5). For SYNOP codes not associated with FZRA, this proportion can be interpreted as the false alarm rate F (A6). Encouragingly, the largest proportions are for codes that are for light FZRA (codes 66, H=26%), and moderate to heavy FZRA (code 67, H=34%). The proportion for FZRA observed during past hour (code 24) is somewhat lower (H=10%) but still non-zero. Other clearly non-zero proportions are for codes of freezing drizzle (codes 56, F=6%, and 57, F=6%) and ice pellets
(code 79, F=13%) that physically resemble FZRA.

### 3.2.2   Number of freezing rain events at weather stations

The annual numbers of FZRA events are reproduced better as spatial averages across the weather stations than at individual sites (Fig. 3 and Table 3). The spatial averages were computed over all the 200 stations in the calibration set and separately for the stations in the coastal, semi-coastal and continental subgroups. $\text{FMI}_{CLIM}$ produces marginally better correspondence to
observations than $\text{FMI}_{NWP}$, when considering the correlation coefficients (0.93 vs. 0.90) between the predicted and observed, spatially-averaged yearly numbers of FZRA cases per station in all stations. $\text{FMI}_{CLIM}$ has a higher correlation also in all subgroups. $\text{FMI}_{NWP}$ reproduces the mean values in subgroups better, while standard deviation, which is generally overestimated





by both algorithms, is slightly better modelled by the $FMI_{CLIM}$ algorithm. The RMS error of climatological mean is equal the standard deviation of observations, and, encouragingly, the RMS error of all spatial averages is smaller than the standard deviation of observations, implying results are better using $FMI_{C}LIM$ than just using the climatology. For individual sites, however, the RMS error is slightly worse than the standard deviation of observations.

The total number of FZRA events at each station according to observations was compared with the number of FZRA events according to $FMI_{CLIM}$ (Fig. 4). The comparisons were performed for all the 200 stations in the calibration set and separately for the three subgroups of the stations. In each case the root mean squared error (RMS) and the mean error (ME) are calculated, and the $FMI_{CLIM}$ was modelled as the function of observations using the local polynomial regression method Loess (Venables and Ripley, 2002). The mean number of events calculated with $FMI_{CLIM}$ is almost the same (ME=0.0) with those observed

for "all stations" (Fig. 4a). However, the distributions are rather different, the distribution of $FMI_{CLIM}$ result is somewhat symmetric around the mean, but in case of observations the number of stations with small number of FZRA events is higher, and the tail of stations with high frequency of events is much longer. For "all stations" and small numbers of events, $FMI_{CLIM}$ models the average number of events well with some overestimation, and the Loess curve is very near the diagonal line. However, for larger values, the Loess curve is nearly horizontal, implying that $FMI_{CLIM}$ cannot model properly the stations

where the large values occur. In the continental group (Fig. 4d), the curve is nearly horizontal for all values. In the coastal group the RMS error is the smallest but there is an overestimation (ME>0), while for the continental group the RMS error is the largest and there is an underestimation (ME < 0). Smaller RMS error in coastal areas can be partly explained by the lack of large values that would contribute to RMS.

Spatially the largest biases at the individual validation stations were found in central Europe (Fig. 1). A coherent, independent

of algorithm, and persistent (similarity between different decades, not shown) area of underestimation was found in southern Germany and in Austria, while overestimation mostly happens in the northern and eastern validation stations. As explained above, two reasons why stations were excluded from the validation were the low observed number of FZRA events and the different definition of observed freezing rain. Most of the stations that were excluded from the validation due to the low observed number of FZRA events are located on coasts. This lack of freezing rain is modelled well, assuming that observations

are correct. On the contrary, the difference between observations and algorithm results are most prominent in the eastern stations, which were excluded from validation due to the different definition of observed freezing rain.

### 3.2.3 Validation of near-surface predictor variables

In addition to the distance to the coastline, orography is likely to affect the occurrence of FZRA (Sect. 2.1). This suggests that highly variable terrain might cause strong local maxima in the observed occurrence, as the fringe areas of large plains surrounded by mountains are more prone to FZRA. For example, the slopes and valleys surrounding the Great Hungarian Plain

might experience cold air damming, a phenomenon which is associated with ice storms in Northern America (Forbes et al., 1987). This phenomenon can also happen in smaller scales, but presumably it cannot be resolved in our work due to the coarse spatial resolution of the reanalysis data. To explore the possible effect of resolution to the accuracy of reanalysis variables, the correlation coefficients between SYNOP observations and ERA-Interim reanalysis surface variables were calculated in the





validation stations. Indeed, correlations for near surface temperature and humidity of low altitude stations (0.81 and 0.65 for temperature and humidity respectively) are higher than in high altitudes (0.52 for temperature, 0.44 for humidity). Presumably, the highest stations usually are located in highly variable terrain, and rather large grid cells of the reanalysis do not represent well the observed variability of these stations. No strong differences in observed and ERA-Interim mean values of the variables

were found, but the differences are again larger in the highest stations, so that ERA-Interim overestimates, for example, the temperatures by $1.6°C$ at higher and only $0.3°C$ at lower altitudes.

### 3.2.4    Vertical profiles of temperature and humidity

Figure 6 shows vertical temperature and humidity profiles of ERA-Interim in the validation stations during the observed and predicted FZRA events. As intended, the $\text{FMI}_{CLIM}$ algorithm picks the cases where well-defined melting and freezing layers

exist (Fig. 6c). However, as the wider variability ranges in Fig. 6a show, FZRA was observed in much more diverse temperature profiles than predicted. The same feature can be seen in the humidity profiles (Fig. 6b, d). The $\text{FMI}_{CLIM}$ algorithm selects cases where the melting layer humidity is high, even though in reality the precipitation can be formed above the melting layer and thus humidity in the melting layer can be low during the FZRA events. The observed and predicted number of FZRA cases (Fig. 6a – e) is 7900, but only 1750 events happened simultaneously in observations and in prediction (Fig. 6e, f).

### 3.2.5    Duration and spatial extent of freezing rain events

The probability of the most common FZRA events, those detected during a single 6-hour time step only, is predicted similarly as in observations by both FMI algorithms (Fig. 7a). For longer-lasting events the algorithms produce overestimates that increase towards the extreme tail of the durations; at the $10^{-6}$ probability level the overestimation is about six hours. The spatial extent of FZRA is also overestimated in the extreme tail (Fig. 7b) so that $\text{FMI}_{NWP}$ overestimates the number of simultaneously

impacted stations by 20% and $\text{FMI}_{CLIM}$ by 50% at the $10^{-4}$ probability level. Additionally the most frequent events, i.e. one impacted station, are slightly underestimated by both algorithms. Durations are modelled better at continental stations than at coastal and semi-coastal stations, and number of impacted stations is modelled almost correctly at coastal stations but poorly at continental stations, with overestimation of almost 100% by $\text{FMI}_{CLIM}$ at the lowest $10^{-5}$ probability level (not shown).

### 3.3    Climatology of freezing rain in Europe

The spatially averaged annual mean numbers of FZRA events in different subgroups varies from 0.8 in coastal stations to 1.3 in continental stations as seen in Table 3. Figure 3 shows that the interannual variability of FZRA events is substantial. Even more importantly, the coefficient of variation – standard deviation divided by mean – is large especially in the continental subgroup: there are years with less than 0.5 FZRA events per station, and years with more than three events on average. Large coefficient of variation may hamper the anticipation and allocation of resources in road maintenance, for example. Weak positive lag-1-

year autocorrelations were found in the annual numbers, ranging from 0.20 in semi-coastal to 0.32 at coastal stations, which indicates weak but non-zero predictability of the annual FZRA number based on the number of the previous year.



The spatial distribution of the annual mean number of events (Fig. 8a) shows that FZRA is the most frequent in eastern Europe. Large areas in Belarus, Ukraine and Russia encounter 3–4 FZRA events per year, locally 4–5 6-hourly events. The maximum annual number of FZRA cases is situated over The Carpathian mountains, where locally over 5 events were found on average. The spatial distribution of maximum durations of the events (Fig. 8b) follow qualitatively the mean occurrence distribution in general, but some areas where FZRA is relatively rare, for example the Benelux countries and the Oslo Fjord, have encountered at least one prolonged event. Almost regardless of the latitude, the coastal and marine areas do not experience FZRA as often as the other regions, apparently because water bodies effectively prevent near-surface cold layers that are essential for the formation of freezing rain. An exception is the northern Baltic Sea, having long ice cover season and relatively frequent FZRA cases.

The FZRA season begins in northern parts of the Fennoscandia already in September, which can be seen in the monthly climatology maps (Fig. 9). In that area the phenomenon is experienced most frequently in November. After that, in December–February, the temperatures drop so low that the melting layer seldom forms. Probably for this reason the total number of FZRA events in northern Europe is rather small, even though the season is the longest, lasting until May. In central and especially in eastern Europe the season is shorter but much more intense, so that the annual number of FZRA events is larger than in the northern parts of the continent.

The most widespread (Fig. 7b, d) FZRA events at the $10^{-4}$ probability level covered over 600 000 km$^2$ and impacted over 10% of the weather stations. The most long-lasting events below the $10^{-7}$ probability level lasted over 50 hours (Fig. 7a, c). It is worth noting that the most long-lasting cases are not necessarily the same as the most widespread events. The proportion of simultaneously impacted stations in the subgroup varies from 13% (coastal) through 16% (semi-coastal) to 30% (continental stations) at the $10^{-4}$ probability level (not shown).

## 4 Discussion

In this paper a freezing rain detection algorithm has been introduced together with a method to calibrate it. After validation the algorithm was applied to a reanalysis in order to construct the European occurrence climatology of freezing rain. This far, no complete gridded climatologies of freezing rain have been presented for Europe in literature. A physically justified, statistically adjusted algorithm which is mainly based on the vertical temperature profile of the atmosphere was used in the study to ensure the credibility of the result. Sub-daily, quality controlled European wide SYNOP weather station data was used in the statistical adjustments. The different definitions of freezing rain in different countries makes derivation of observation-based climatology of freezing rain difficult. The gridded climatology is thus more homogeneous compared to climatologies based only on weather station data.

In validation, the gridded meteorological dataset is compared with the point-like surface observations. It is possible that in some cases ERA-Interim represents the occurrence of FZRA inside the 0.7° grid cells better than the stations. Each grid cell represents the mean state of variables inside the cell, which by definition can not be the same as in the weather stations representing very local variability of the atmosphere.





### 4.1 Possible sources of uncertainty

Possible sources of uncertainty in the gridded climatology of freezing rain in Europe, besides the detecting algorithm itself, include human errors in observing FZRA, deficiencies in the ERA-Interim reanalysis data, effects of subgrid-scale orography. These issues are discussed in more detail in the following.

Observing FZRA correctly remains a challenge for observers, as the phenomenon can be easily confused with ordinary, non-freezing rain. Particularly minor cases are difficult to detect for two reasons: firstly, they do not necessarily cause significant ice accretion on structures, which would help the identification of FZRA, and secondly, confusion of FZRA with freezing drizzle or ice pellets might happen, especially by inexperienced observers. Additionally, minor cases might not be recorded in 6-hourly time interval due to their short duration (Ressler et al., 2012; Cortinas, 2000), which adds uncertainty to the observations of
this study.

To maximize the reliability of the observations, a large number of SYNOP stations was used, which is believed to average out random errors. Additionally the stations having the most complete time series and regular, high-frequency manual observations were included. The strongest observational biases were identified at the eastern stations (as discussed in Sect. 2.1 and as seen in Fig. 1), and for that reason they were excluded from the study. Still, the effect of human errors can not be totally removed
by applying selection techniques to the existing observations.

Besides the observational uncertainty, a potential source of uncertainty arises from the ERA-Interim reanalysis data. The low-level wintertime temperature inversions in the data are known to be lacking at Arctic latitudes as shown by Serreze et al. (2012). Arguably their result might be valid at least to some extent outside the Arctic. This uncertainty was considered by calibrating the original $FMI_{NWP}$ algorithm instead of using it as such, but apparently the impact of the bias can not be fully
compensated by simple adjustments of the threshold values in $FMI_{CLIM}$. In our analysis the mean temperature profile (red line in Fig. 6a) of the ERA-Interim reanalysis do not show a clear near-surface freezing layer below the melting layer, which either indicates problems in the above-surface temperatures of the reanalysis, or highlights the importance of the warm rain formation mechanism of FZRA compared to the melting layer – cold layer mechanism; $FMI_{CLIM}$ is not able to see the former cases.

Locally, near the mountainous regions, a potentially major source of uncertainty is caused by the orography, which is strongly smoothed in the $0.7°$ resolution of the ERA-Interim. The results may be especially biased in sub-grid scale valleys, where prolonged FZRA events might be caused by trapped cold air mass. It is possible that the optimal value of $h_{cold}^{thr}$, found here for $FMI_{CLIM}$, differs from the corresponding uncalibrated value because high-elevation or mountainous stations were included in the calibration, and the original, uncalibrated version is mostly used to predict FZRA over the mostly flat-terrain Finland,
where smaller $h_{cold}^{thr}$ might work well or well enough.

In the algorithm, two thresholds were used to detect situations favouring subcooling of raindrops: minimum required depth and the maximum allowed temperature of the near-surface cold layer. However, fulfilling both criteria mentioned above does not totally guarantee the liquid phase: a too cold or a too deep cold layer refreezes the hydrometeors, which is not taken into



account in the current version of the algorithm. This could explain, at least partly, the occasional misclassifications to ice pellets.

## 4.2 Future work

Further exploration of existing data, i.e. observations and the reanalysis, is needed to deepen the knowledge of the phenomenon,
including synoptic analysis of the most extreme cases, calculating the precipitation amounts, and studies of other freezing phenomena, i.e., freezing drizzle and ice pellets. Furthermore, impact assessments taking into account the related factors such as the effect of wind (accelerating the accumulation of ice and straining structures by itself), would be informative. In addition to these, the following are the key issues to improve the credibility of the current occurrence results:

- The description of the near-surface cold layer in the $FMI_{NWP}$ algorithm could be enhanced by defining a closed range
where the parameter $h_{cold}^{thr}$ (or $T_{cold}^{thr}$) should hit, instead of considering lower (or upper) limits alone. Additionally, separating the moist and melting layers to be independent from each other should be tested. Increasing the vertical resolution would be helpful, as small differences in vertical layers easily affect the result (Stewart et al., 2015).

- The optimal parameter values could be derived from different calibration periods to have a set of differently calibrated FMI algorithms. The results could then be used to create an ensemble of FZRA climatologies for the purpose of uncer-
tainty analyses or sensitivity analyses.

- Small uncertainties in the location, in space and time, of the moving precipitation patterns in the reanalysis increase the uncertainty of algorithm-based FZRA detection because of the typical short duration of the FZRA events. To some extent, this uncertainty could be studied by taking into account the preceding and following time steps in the observational records, as the original SYNOP data is 3-hourly. Additionally, preliminary analyses (not shown) indicate that 1-day or
5-day averaging could also enhance the correlation of the results with similarly averaged SYNOP observations.

- A more accurate division between the eastern and other European SYNOP stations, following the borders of the countries, would enhance the calibration and validation processes.

- In addition to FMI algorithms, new identification methodology could be developed or adopted and tested, including statistical classification methods and more complex but well-performing physical methods, such as ones which explicitly
simulate the melting and freezing of descending hydrometeors (e.g., Ramer, 1993).

As shown by Serreze et al. (2012), current reanalyses typically have problems in representing wintertime temperature inversions. Therefore, new predictor datasets need to be tested when available, for example the ERA5 of the ECMWF, which is designed to be the successor of ERA-Interim. The most important criterion when selecting new predictor data would be the accuracy of the vertical temperature structure and high temporal and spatial resolutions, since the temperature profile needs to
be correct for reliable detection of FZRA. Additional observational in situ datasets, such as METAR aviation weather reports and atmospheric soundings, could be used in further development of the FMI algorithm.



As discussed above, neither the observations of FZRA nor methods to predict it are perfect, that is, the ground truth is missing. The methods of then estimating the real base rate of a phenomenon and the verification results of detection are much discussed in social and medical sciences (see, e.g. Lewis and Torgerson, 2012), but are little-known in atmospheric sciences (see Hyvärinen et al. (2015) for the first steps). Ideally, these methods require more than two independent sources of data, for example, different observations and method results. For FZRA this requirement can be difficult to fulfill, as there are not many different sources of observations available and methods are usually developed using all available observations. However, exploring these methods would contribute to the better estimation of the occurrence of FZRA.

## 5 Conclusions

A method for detecting FZRA in gridded, temporally dense meteorological datasets is presented, followed by a climatological European wide mapping for the occurrence of FZRA. The objective of this paper was to develop an algorithm that is simple enough to avoid memory-intensive computation in the analysis of gridded datasets, and on the other hand is physically sensible and sophisticated enough to model the complicated conditions leading to FZRA. The low validation results at station locations indicated that uncertainties related to observations, to the identification method, and to the temporal and spatial resolution of the reanalysis, deteriorate the algorithm-based identification of FZRA events. However, it is not clear which uncertainties are the most important, and it is likely that their relative importance varies in space and even in time.

The freezing rain detection algorithm selected for this study was originally developed in numerical weather prediction. The physically motivated internal thresholds of the algorithm were calibrated using the ERA-Interim reanalysis and SYNOP weather station observations. Values of the thresholds did not change considerably in the calibration process, and the simple calibration did not reveal strong biases in the reanalysis, showing that the original thresholds are already adequate for climatological analysis of freezing rain in ERA-Interim.

According to the algorithm-based analysis of the gridded reanalysis data, freezing rain is more common in central and eastern Europe than in the northern parts and over the coastal regions. The FZRA season begins in September and lasts until May in northern Europe. In central and eastern Europe the season is shorter, beginning in October and lasting until April, but much more intense, leading to more yearly events (typically 2-3 events/year) than in northern countries (typically 0.5-2 events/year). In 1979–2014, the most long-lasting FZRA events lasted over 50 hours and the most widespread events covered over 600 000 km$^2$ in Europe. The overall probability of a 50-hour event is $10^{-7} - 10^{-8}$.

Spatially and temporally coherent information about occurrence of FZRA in Europe has been lacking thus far. The gridded output of this study is a preliminary approach to answer this demand, and as such, the current work can be used as a basis for risk analyses, if the underlying uncertainties are carefully kept in mind. For example, questions such as what year contained the most freezing rain events in continental regions of Europe, can be answered reliably using the current data and method. Station scale analyses, however, require further studies.





## 6 Data availability

Members of the ECMWF can access the MARS archive for the SYNOP weather station data used in this study. ERA-Interim reanalysis data can be obtained from the public server of the ECMWF. Processed data files and Python code are available on request from the corresponding author.

## Appendix A: Verification measures

Results of the comparison of two binary data sources can be presented in a $2 \times 2$ contingency table (Table A1). If one of these data sources represents the true values, the cells can be named as follows: $a$ the number of hits, $b$ false alarms, $c$ misses and $d$ correct rejections. In this study, these two data sources are the SYNOP observations and algorithmic classifications of FZRA, and the true values are SYNOP observations. The terminology follows Jolliffe and Stephenson (2012).

The simple measure of performance is Proportion Correct (PC), defined as

$$PC = \frac{a+d}{a+b+c+d}. \tag{A1}$$

The Critical Success Index (CSI) is similar, but ignores the cell $d$:

$$CSI = \frac{a}{a+b+c}. \tag{A2}$$

Many different skill scores have been developed and in this study two of them are used: the Heidke Skill Score (HSS),

$$HSS = \frac{2(ad-bc)}{(a+c)(c+d)+(a+b)(b+d)}, \tag{A3}$$

and the Symmetric Extremal Dependence Index (SEDI),

$$SEDI = \frac{\ln F - \ln H + \ln(1-H) - \ln(1-F)}{\ln F + \ln H + \ln(1-H) + \ln(1-F)}, \tag{A4}$$

where the hit rate (H), the ratio of correct FZRA classifications to the number of times the FZRA weather code was observed, is

$$H = \frac{a}{a+c}, \tag{A5}$$

and the false alarm rate (F), the ratio of false FZRA classifications to the number of times the FZRA weather code was *not* observed, is

$$F = \frac{b}{b+d}. \tag{A6}$$

Finally, the bias is defined as the ratio of FZRA classifications to the number of times the FZRA weather code was observed

$$B = \frac{a+b}{a+c}. \tag{A7}$$





*Author contributions.* Matti Kämäräinen collected and processed the data of this study, designed and implemented the optimization and validation procedures in co-operation with Otto Hyvärinen, and prepared the manuscript with contributions from all coauthors. Otto Hyväri-nen designed the cross-validation and validated the classification against SYNOP present weather codes and participated in writing. Simo Neiglick developed the original FMI$_{NWP}$ algorithm used in weather predictions. Jaakko Nuottokari compared ERA-Interim and SYNOP

5    observations in station level. Andrea Vajda, Hilppa Gregow, and Kirsti Jylhä helped with the science, wrote parts of Introduction and commented on the manuscript.

*Acknowledgements.* This work was partly funded by the European Union's Seventh Programme for research, technological development and demonstration under the RAIN project (Risk Analysis of Infrastructure Networks in response to extreme weather; http://rain-project.eu/; grant agreement N° 608166). The work has also received funding from the Ministry of Employment and the Economy in Finland and from

10   the Swedish Radiation Safety Authority through the EXWE project (Extreme weather and nuclear power plants) of the SAFIR2018 program (The Finnish Nuclear Power Plant Safety Research Programme 2015-2018; http://safir2018.vtt.fi).

We thank Sami Niemelä for helping with MARS data retrieval, Tiina Ervasti and Curtis Wood for improving the grammar of the article, and Juulia Lahdenperä for commenting the text. Some of the results were presented in annual meeting of the European Meteorological Society (EMS) in September 2015 by Kämäräinen et al. (2015).





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



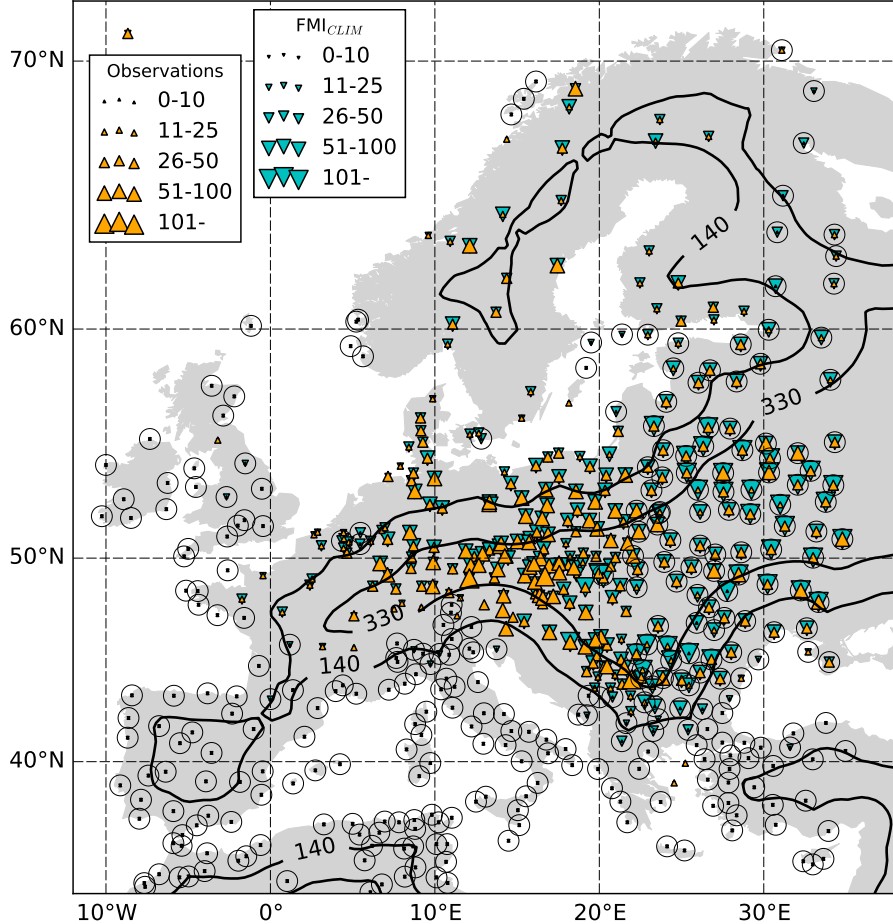

**Figure 1.** Total number of FZRA cases 1979–2014 in Europe according to 6-hourly SYNOP observations (orange) and the $FMI_{CLIM}$ algorithm (cyan), applied to the 6-hourly ERA-Interim reanalysis. The size of the markings indicates the frequency classes. The distance to the nearest coastline (in km) is shown with black isolines, which divide the stations into coastal, semi-coastal and continental groups. Stations that were excluded from the calibration, validation and further analyses of the $FMI_{CLIM}$ algorithm, are indicated with circles.

```
pLevels = (925, 850, 700)
for all timesteps and stations/gridcells :
        T_cold = T2m
        T_max = max (T_925 , T_850 , T_700)
        if (T_cold ≤ T_cold^thr and T_max > T_melt^thr and Pr > Pr^thr) :
                p_cold = nearest pressure level (among pLevels) above (p_surf − h_cold^thr)
                moistMeltLayer = any (T [i] > T_melt^thr and RH [i] ≥ RH_melt^thr) ;  i  = [pLevels ≤ p_cold]
                if moistMeltLayer exists : FZRA = True
                else : FZRA = False
        else : FZRA = False
```

**Figure 2.** A pseudocode representation of the FMI algorithm. See text for definitions of symbols and for description of the logic.




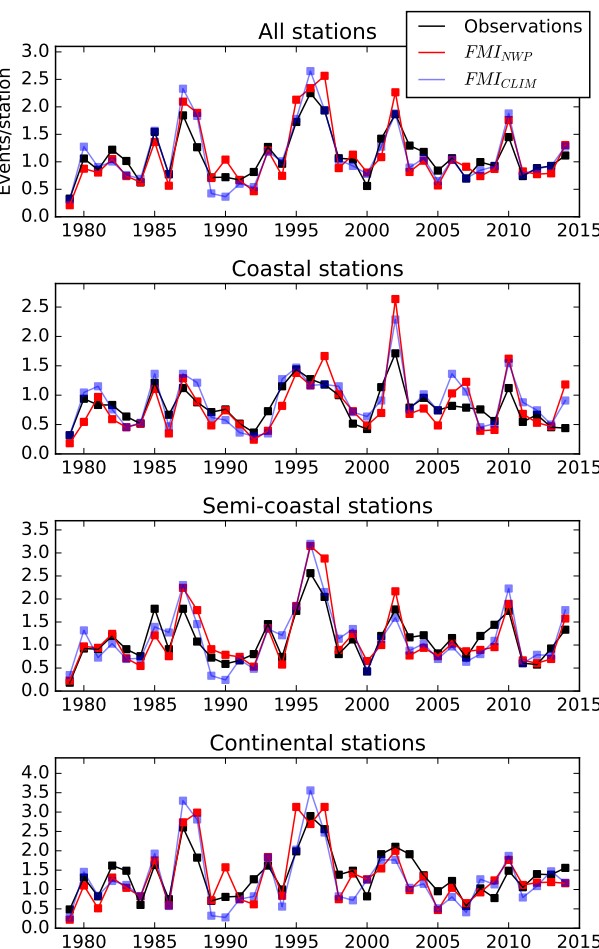

**Figure 3.** Annual, spatially averaged mean number of FZRA cases per station in all 200 stations and in groups based on distance to the nearest coastline according to SYNOP observations (black), the $FMI_{NWP}$ (red), and the $FMI_{CLIM}$ (blue) algorithm. Definitions of groups can be seen in Fig. 1. Statistics calculated from the numbers presented here are shown in Table 3. Note the different y-axis scales.




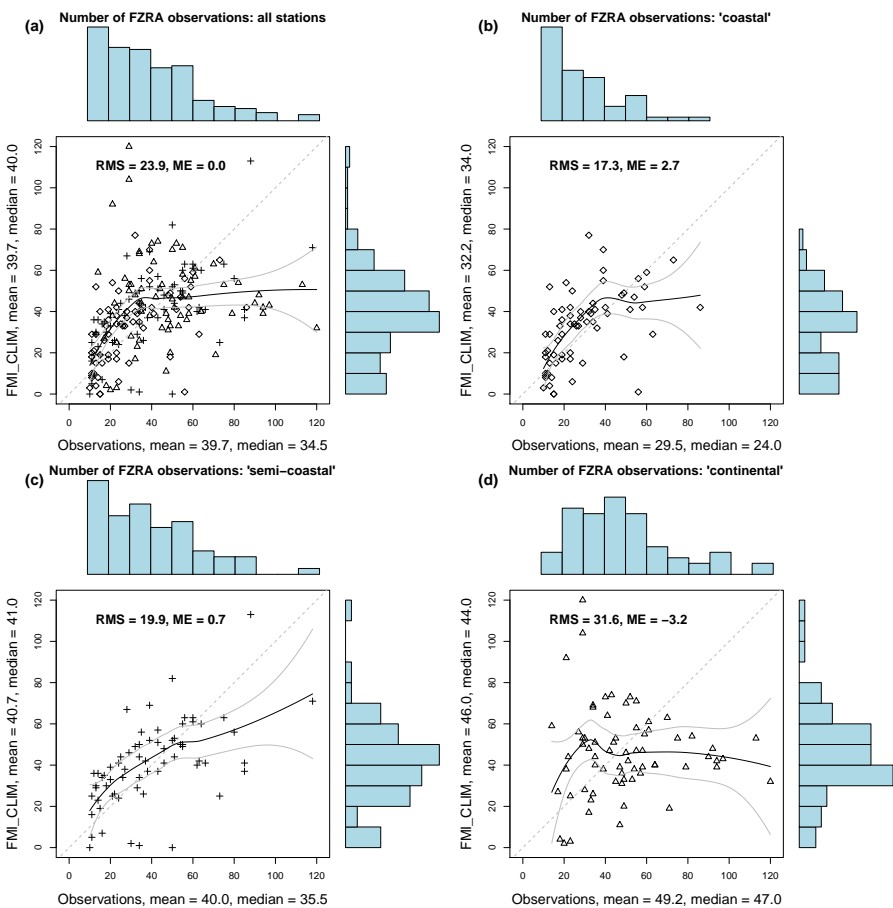

**Figure 4.** The total number of FZRA events according to observations compared with the number of FZRA events according to FMI$_{CLIM}$ for stations in the calibration set, using (a) all 200 stations and stations in (b) coastal, (c) semi-coastal and (d) continental groups. The curves superimposed in the scatter plots show FMI$_{CLIM}$ as the function of observations using the Loess method (Venables and Ripley, 2002) together with the 95% confidence interval. Note that plotting symbols for groups in (b), (c), and (d) are used also in (a). The bar diagrams present relative frequency distributions.





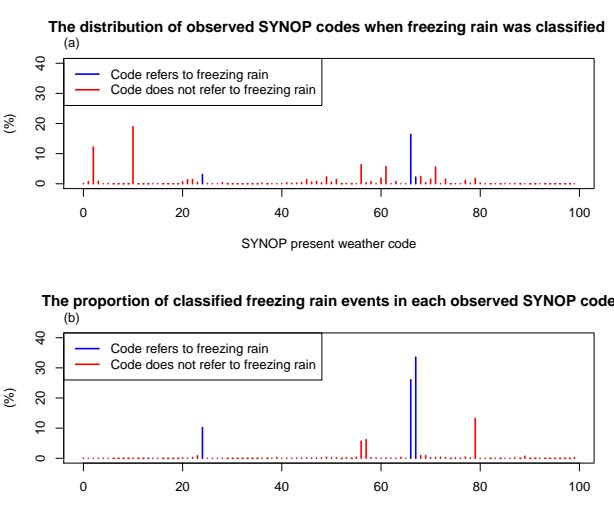

**Figure 5.** (a) The distribution of observed SYNOP present weather codes when an event was classified as FZRA by the FMI$_{CLIM}$ algorithm. The distribution sums up to 100%. (b) The proportion of cases classified as FZRA in each observed SYNOP present weather code. Each code can have a proportion from 0% to 100%.




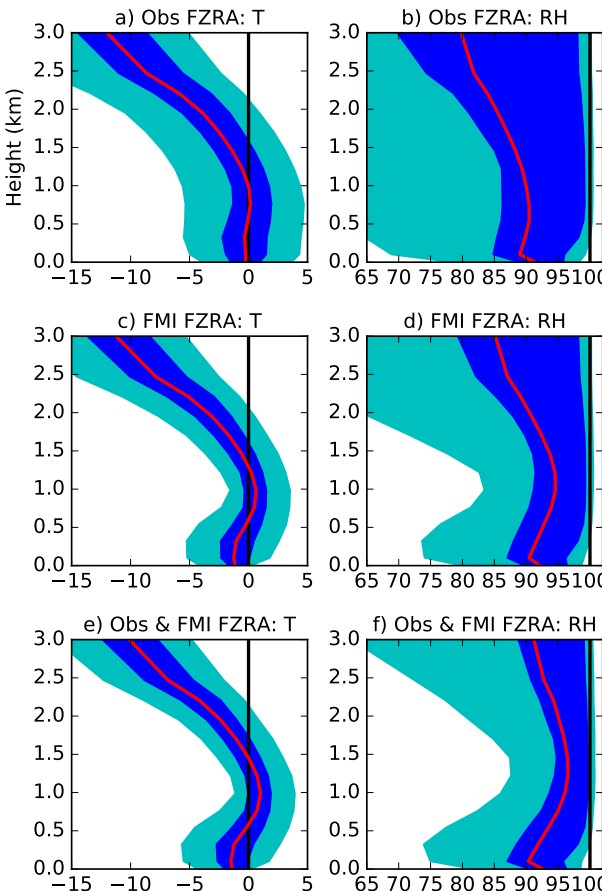

**Figure 6.** Vertical profiles of temperature (°C, left column) and relative humidity (%, right column) of ERA-Interim at weather station locations. 5%–95% range (cyan), 25%–75% range (blue) and mean (red) are shown. Top row: profiles when FZRA was reported in SYNOP messages (7900 events in total). Middle row: FZRA profiles according to the calibrated $\text{FMI}_{CLIM}$ algorithm (7900 events). Bottom row: profiles where both the $\text{FMI}_{CLIM}$ algorithm and observations indicated FZRA (1750 events).




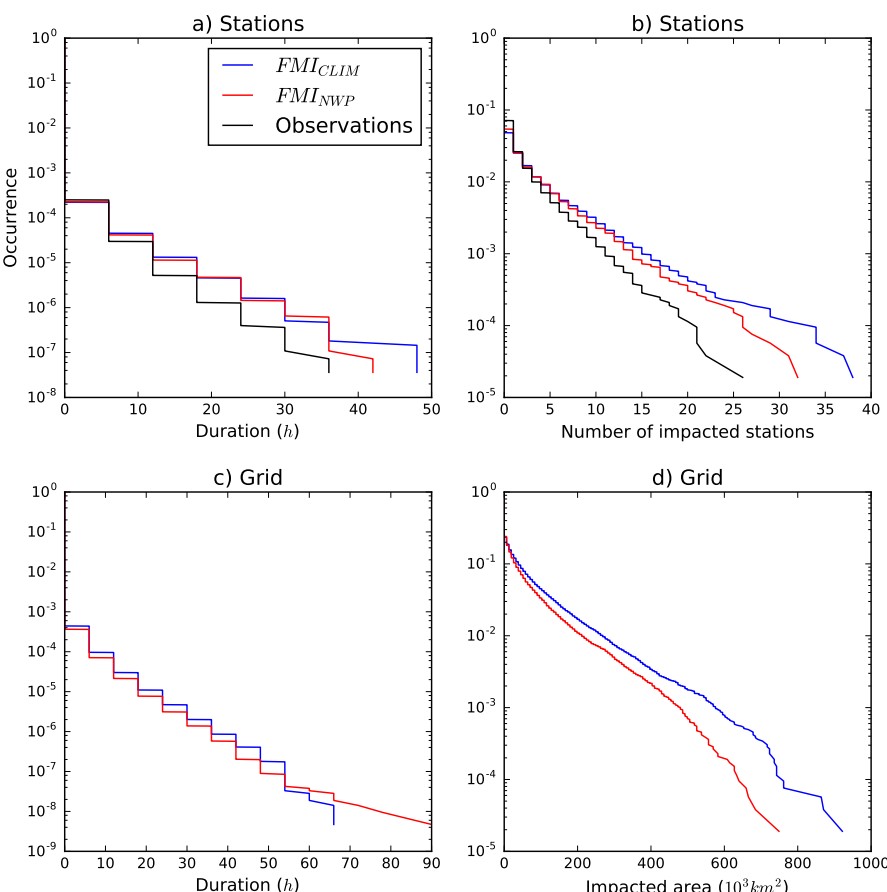

**Figure 7.** Probability of duration (left column) and spatial extent (right column) of FZRA events at station locations (top row) and in all grid cells (bottom row) according to the detection algorithms (blue, red) and observations (black).





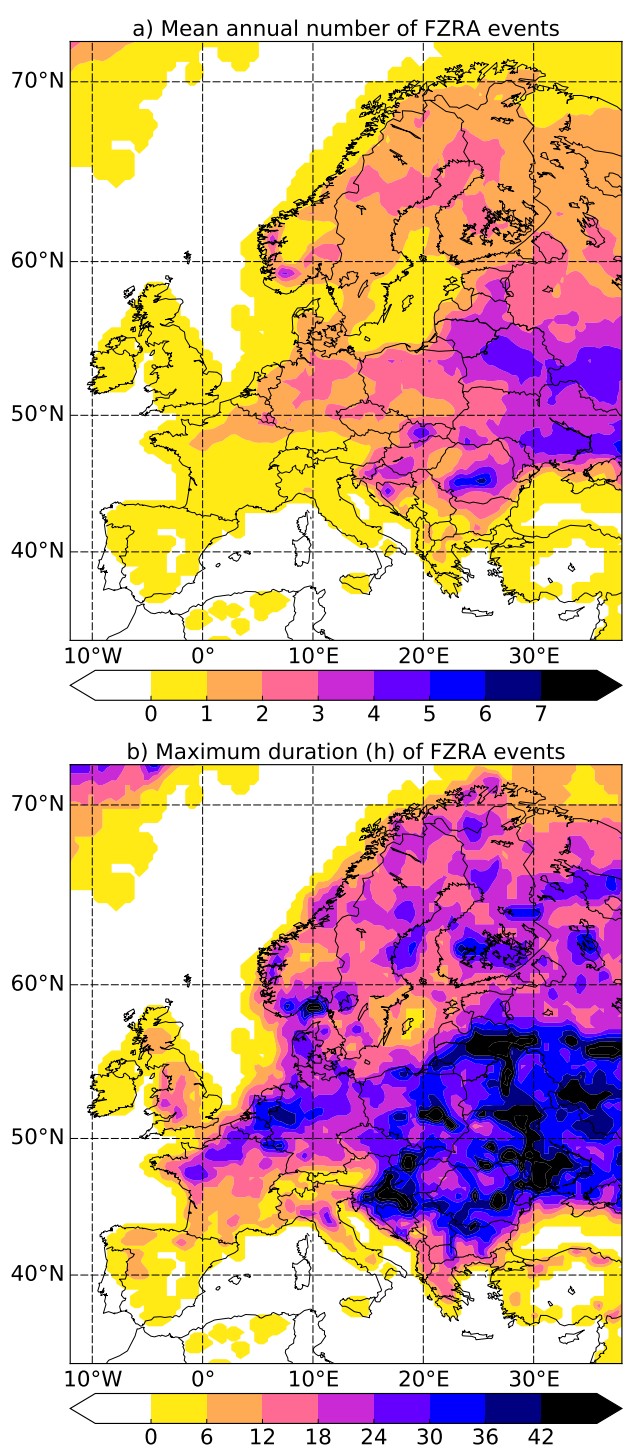

**Figure 8.** (a) Mean annual number of FZRA events and (b) maximum duration of events in the 1979-2014 study period. FMI$_{CLIM}$ algorithm is applied to the ERA-Interim reanalysis data.





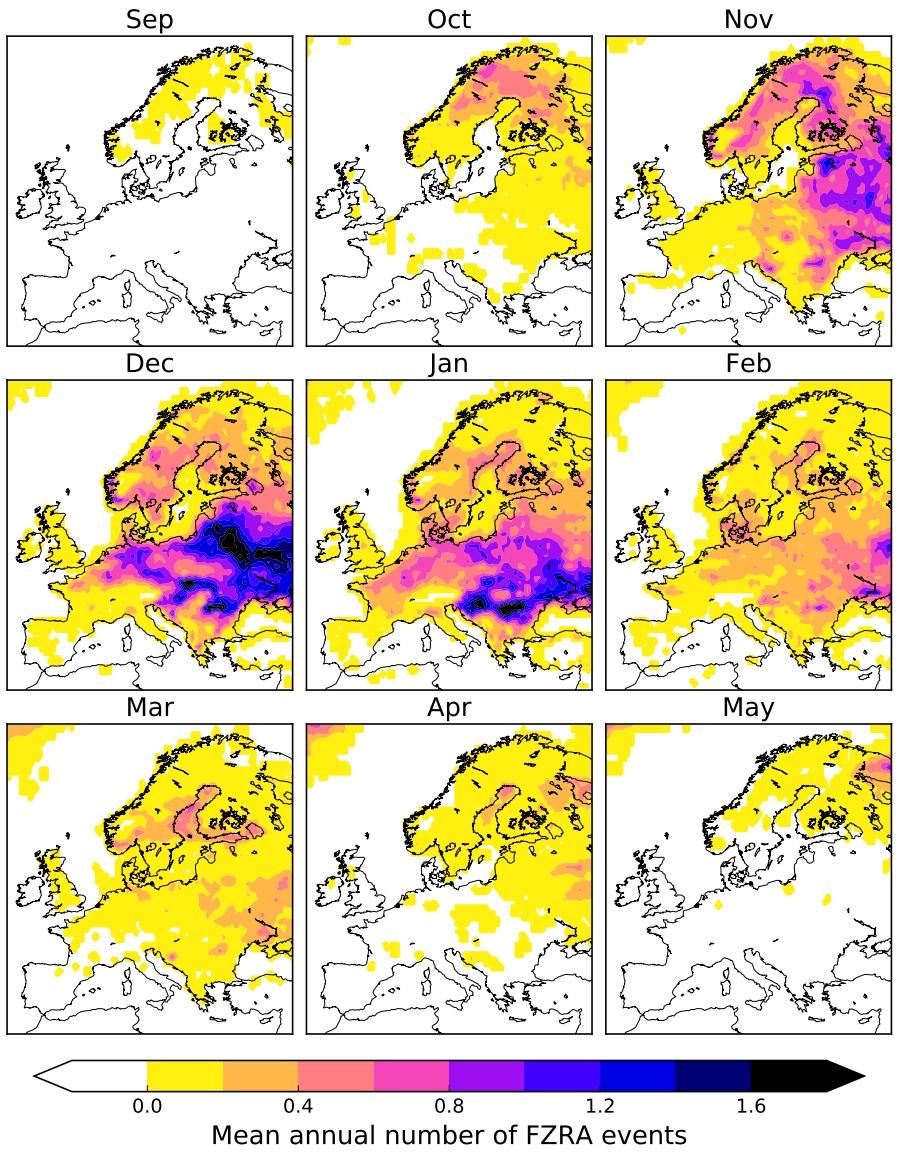

**Figure 9.** The monthly climatology of FZRA in 1979–2014 according to the $FMI_{CLIM}$ algorithm, applied to the ERA-Interim. The average annual number of 6-hourly FZRA cases is shown.





**Table 1.** Uncalibrated (upper row) and calibrated (bottom), optimal values of threshold parameters in the 29-year calibration periods. Mean values of the optimal values are shown, computed for calibration periods using the sample variance of period values. Mean values are used in the final analysis of the gridded dataset. $h_{cold}^{thr}$ = minimum cold layer depth; $RH_{melt}^{thr}$ and $T_{melt}^{thr}$ = minimum humidity and minimum temperature in the melting layer; $T_{cold}^{thr}$ = maximum cold layer temperature; and $Pr^{thr}$ = minimum surface precipitation rate.

| | $h_{cold}^{thr}$ (hPa) > | $RH_{melt}^{thr}$ (%) > | $T_{melt}^{thr}$ (°C) > | $T_{cold}^{thr}$ (°C) < | $Pr^{thr}$ (mm 6h$^{-1}$) > |
|---|---|---|---|---|---|
| FMI$_{NWP}$ | 15 | 90 | 0 | 0 | 0.04 |
| FMI$_{CLIM}$ | 65 | 86 | -0.58 | 0.32 | 0.32 |

**Table 2.** The cross-validation measures and scores in 7-year validation periods when predicted 6-hourly FZRA result is compared against observed 6-hourly events. Mean values and standard errors, computed for validation periods using the sample variance of period values, are shown. See text and Appendix A for definitions of measures and scores.

| | FMI$_{NWP}$ | FMI$_{CLIM}$ | FMI$_{CLIM}$ - FMI$_{NWP}$ |
|---|---|---|---|
| CSI | $0.103 \pm 0.005$ | $0.123 \pm 0.007$ | $0.021 \pm 0.004$ |
| SEDI | $0.632 \pm 0.008$ | $0.664 \pm 0.013$ | $0.032 \pm 0.006$ |
| HSS | $0.185 \pm 0.008$ | $0.219 \pm 0.012$ | $0.034 \pm 0.006$ |
| $a$ | $290 \pm 20$ | $340 \pm 20$ | $47 \pm 11$ |
| $b$ | $1290 \pm 120$ | $1190 \pm 80$ | $-100 \pm 50$ |
| $c$ | $1280 \pm 70$ | $1230 \pm 80$ | $-47 \pm 11$ |
| H | $0.186 \pm 0.009$ | $0.217 \pm 0.015$ | $0.032 \pm 0.008$ |
| F | $0.00069 \pm 0.00005$ | $0.00064 \pm 0.00004$ | $-0.00005 \pm 0.00003$ |
| B | $1.00 \pm 0.05$ | $0.97 \pm 0.04$ | $-0.02 \pm 0.04$ |





**Table 3.** Statistics calculated from numbers presented in Fig. 3. Correlation coefficient of algorithm results compared to observations ($r$), mean value ($\overline{x}$), standard deviation ($s$), and RMS error (RMS) of annual mean numbers of FZRA cases per station averaged over all stations, averaged over groups based on distance to the nearest coastline, and in individual stations are shown.

| | $\text{FMI}_{NWP}$ | | | | $\text{FMI}_{CLIM}$ | | | | Observations | |
| --- | --- | --- | --- | --- | --- | --- | --- | --- | --- | --- |
| | $r$ | $\overline{x}$ | $s$ | RMS | $r$ | $\overline{x}$ | $s$ | RMS | $\overline{x}$ | $s$ |
| All stations | 0.90 | 1.10 | 0.57 | 0.26 | 0.93 | 1.10 | 0.54 | 0.22 | 1.10 | 0.42 |
| Coastal | 0.78 | 0.81 | 0.48 | 0.31 | 0.85 | 0.89 | 0.43 | 0.24 | 0.82 | 0.31 |
| Semi-coastal | 0.89 | 1.14 | 0.65 | 0.31 | 0.89 | 1.13 | 0.63 | 0.30 | 1.11 | 0.49 |
| Continental | 0.81 | 1.34 | 0.76 | 0.45 | 0.86 | 1.28 | 0.79 | 0.42 | 1.37 | 0.58 |
| Individual stations | 0.40 | 1.10 | 1.61 | 1.70 | 0.44 | 1.10 | 1.59 | 1.62 | 1.10 | 1.48 |

**Table A1.** Contingency table of the comparison between observations and the algorithm. The symbols $a$-$d$ represent the different number of FZRA events observed to occur in each category.

| | Observation | |
| --- | --- | --- |
| Algorithm | Freezing rain | No freezing rain |
| Freezing rain | $a$ (Hit) | $b$ (False Alarm) |
| No freezing rain | $c$ (Miss) | $d$ (Correct Rejection) |