# Peer review of "A method to estimate freezing rain climatology from ERA-Interim reanalysis over Europe"

_Natural Hazards and Earth System Sciences, 2016_

## Referee Comment (RC1) · D. Nikolov (Referee) · 20 Sep 2016

Referee comment on the paper "A method to estimate freezing rain climatology from ERA-Interim reanalysis over Europe" with authors M. Kämäräinen et al.

Dimitar Nikolov National Institute of Meteorology and Hydrology 1784 Sofia, Bulgaria

General comments The paper describes a method for estimation of the freezing rain (FZRA) climatology from the ERA-Interim reanalysis. The method is based on an algorithm for determination of the precipitation type, which uses the vertical profiles of relative humidity and air temperature. The algorithm is briefly described and then the results from its evaluation and calibration with SYNOP weather station observations are presented. Finally, the method is applied on gridded data from the ERA-Interim reanalysis for estimation of some characteristics of the freezing rain events in Europe

such as mean annual and monthly numbers, duration and spatial extent.

The paper is structured in 5 paragraphs as follow: a pertinent abstract with briefly and clearly summary of the work done; an introduction with brief description of the previous and related work in this field, the topicality of the problem and the data and methods used; a paragraph with description of these data and methods, followed by a paragraph with the results and the main body of the manuscript concludes with two paragraphs for discussion and conclusions. The availability of the data is given in paragraph 6 and the verification measures, used in paragraph 2, are presented in a separate annex.

The manuscript represents a substantial contribution to the climatology of freezing rains in Europe. As the authors correctly point out, despite the severe impact of this phenomenon, the publications on its climatology for the European region are quite few in contrast to these for North America. Thus the proposed publication is timely. The used approach for estimating of the FZRA climatology, based on the application of the proposed evaluated and calibrated method on gridded meteorological data, is appropriate and ensures spatial and temporal cohesion of the results, despite some uncertainties.

The description of the data and the methods used is sufficiently complete and accurate and allows reproduction.

The consideration of previous and related work is also sufficient and the cited references are appropriate.

The conducted research is sound and comprehensive and the achieved results are discussed in an appropriate and balanced way.

Specific comments

I would recommend the authors replace the references Rauber et al., 2000; Carrière et al., 2000 for the warm rain process (page 2) with (or add) the following two:

Bocchieri, J., 1980: The objective use of upper air soundings to specify precipitation

type. Mon. Wea. Rev., 108, 596–603. Huffman, G. J., and G. A. Norman Jr., 1988: The supercooled warm rain process and the specification of freezing precipitation. Mon. Wea. Rev., 116, 2172–2182.

I disagree with the decision of the authors to exclude most of the stations from eastern Europe. It seems that they have erroneously interpreted the explanations of Bezrukova et al. (2006) for the different definitions of FZRA events in these countries. Indeed, sometimes the icing due to supercooled clouds or fogs may deposit as glaze (wet growth process) and then the symbol for glaze is written down, but such a case will never be reported as freezing rain or freezing drizzle in the WMO weather codes. This ambiguity concerns mostly the local meteorological archives where additional control is needed to distinguish between both events. The weather codes 24, 56, 57, 66 and 67 are not affected at all. By this reason the authors (of Bezrukova et al., 2006) have decided to restrict only to the WMO codes.

The authors of the manuscript have also filtered the data outside the interval –30oC Ãů + 10oC, which seems to be too wide. Most often FZRA occur in the interval –10oC Ãů 0oC, so an appropriate interval for filtering, in my opinion, would be –15oC Ãů + 5oC. This would prevent to a certain extent from misclassification of ice pellets as FZRA or FZDR.

The finding that the altitude does not contribute to the explain variance is somehow surprising for me. One would expect that the number of FZRA and their duration would decrease with the altitude because of the decreasing of the depth of the near-surface cold layer and FZRA aloft should be even more rare event than the FZRA at the ground. However, mountain ranges mostly caused cold air damming which is difficult to be recognized in data sets with coarse resolution.

The vertical resolution of the FMINWP seems to be not very appropriate for detailed representation of the vertical profiles of the relative humidity and the air temperature, which would affect the correct estimation of the near-surface cold layer and the melting

layer above. It can be seen that an increasing of the resolution is foreseen as future work and this would be very helpful.

The minimum acceptable cold layer depth has been significantly increased by the calibration procedure – from 130 meters up to 400 meters. This seems very reasonable because of the large size of the investigated area and variable weather conditions. For example Bernstein reported values of the near-surface cold layer in USA between 100 and 1400 meters, the minimums being between 100 and 300 meters (Bernstein, B., 2000: Regional and local influences on freezing drizzle, freezing rain, and ice pellet events. Wea. Forecasting, 15, 485–508.).

The annual numbers of the FZRA events is indeed well reproduced by the algorithm (fig. 3), but its hit rate of 20% and the behaviour of its results for large number of events (fig. 4) are another indication for the need of future improvement of FMICLIM.

Both algorithms predict quite well the most typical duration of the FZRA events but overestimate the longer-lasting events as well as the events with large spatial extent.

Very interesting results are presented in the paragraph 3.3 Climatology of freezing rain in Europe. However, the finding for a maximum in the annual number of events over the Carpathian mountain sounds surprisingly for me. It would be useful if the altitude of these regions is given.

Technical comments I have encountered only two small misprints – on page 9, third row – the FMICLIM is written wrongly and on page 11, third row is written "The Carpathian . . .".

Conclusions The paper addresses relevant scientific questions regarding the climatology of FZRA for the European region which are within the scope of NHESS.

The paper presents new data and new coherent results for FZRA occurrence in Europe and all these are up to international standards.

The scientific methods and assumptions are valid and outlined clearly and the results

are sufficient to support the interpretations and the conclusions.

The authors have reached substantial conclusions about the spatial and temporal distribution of the main characteristics of the FZRA in Europe – mean monthly and annual number of cases, duration of the FZRA events and their spatial extent.

The description of the used data and methods, as well as of the obtained results, is sufficiently complete and accurate and allows their reproduction by fellow scientists.

The title clearly and unambiguously reflects the contents of the paper.

The abstract provides a concise, complete and unambiguous summary of the work done and the results obtained.

The title and the abstract are pertinent, and easy to understand to a wide and diversified audience.

The mathematical formulae, symbols, abbreviations and units are correctly defined, described and used. The formulae, symbols and abbreviations are numerous and there is an appendix listing them.

The size, quality and readability of each figure is adequate to the type and quantity of data presented, except for figure 5 which could be a little bit larger.

The authors give proper credit to previous and related work with a small oversight of two references for the warm rain process.

The authors have indicated clearly their own contribution.

The number and quality of the references are appropriate taking into account the mentioned small oversight.

The references are accessible by fellow scientists.

The overall presentation is well structured, clear and easy to understand by a wide and general audience.

The length of the paper is adequate.

There is not any part of the paper (title, abstract, main text, formulae, symbols, figures and their captions, tables, list of references, appendixes) that needs to be clarified, reduced, added, combined, or eliminated. The authors should only take into account that they could use for future investigation the stations in eastern Europe with no restrictions, as far as they utilize the international weather WMO codes.

The technical language is precise and understandable by fellow scientists. The English language is of good quality, fluent, simple and easy to read and understand by a wide and diversified audience.

The amount and quality of supplementary material (one annex) is appropriate.
* * *

---

## Referee Comment (RC2) · Anonymous Referee #2 · 3 Oct 2016

Review: A method to estimate freezing rain climatology from ERA-Interim reanalysis over Europe By: M. Kämäräinen et al.

General Comments:

This article represents an attempt to produce and examine a gridded dataset of freezing rain over Europe as well as to examine this issue with station information. The study makes a number of assumptions regarding the conditions leading to freezing rain although it does end up with a gridded product along with some analysis. There are several issues associated with this article as identified below. Such issues need to be addressed before the article is acceptable.

The article is, in general, structured well and is reasonably well written.

Specific Comments:

Page 1, Line 17: There are many instances of short but 'heavy' freezing rain.

Page 2, Line 12: A recent climatology over parts of northern Eurasia has been completed: Groisman, P.Ya., O. N. Bulygina, X. Yin, R. S. Vose, S. K. Gulev, I. Hanssen-Bauer, and E. Førland 2016: Recent changes in the frequency of freezing precipitation in North America and Northern Eurasia. Environ. Res. Lett. 11, 045007.

Page 3, Line 27: 3-hourly reports are probably insufficient. Most freezing rain events occur at shorter time scales. Was any attempt made to at least 'estimate' how many events were uncounted by using hourly information as well?

Page 3, Line 27: I may have missed this but how did you treat combinations of precipitation types? It is common for freezing rain to occur with ice pellets for example.

Page 4, Line 3: A threshold of 80% is quite low. Why didn't you show the fraction of missing data during the cold season?

Page 4, Line 15: What fraction of observations was beyond these thresholds? Were any of the high valued temperatures associated with very low relative humidities that would lead to much lower wet bulb temperatures?

Page 5, Line 7: This scale is very large for freezing rain. It is quite common for these regions to be less.

Page 6, Line 3: Ice-initiated precipitation is not initially generated in the inversion aloft. What is the implication of assuming it is?

Page 11, Line 31: This paragraph is poorly worded and hard to follow.

Page 12, Line 5: There are standard observing practices to identify freezing rain. Why is this so hard to do?

Page 12, Line 8: You are associating 'minor' with short duration. On what basis? There

can be severe impacts with durations smaller than 6 h and precipitation rates can be high as well..

Page 12, Line 11: Are the errors 'random'?

Page 12 Line 25: Given the enormous smoothing at 70 km, maybe the authors should only consider analyses over 'flat regions'?

Page 12, Line 33: "Occasional misclassification"? How often did this occur?

Page 13, Line 3: To me, this section is too long and wordy. This is a long shopping list. What are the most important and feasible next steps? From my perspective, some of these should be done within this article.

As well, a recent article (Liu et al., 2016) pointed out that precipitation at the surface (including freezing rain) is calculated directly from the model's microphysical package without needing the approach used here. Isn't that the best way forward?

Liu et al., 2016: Continental‑scale convection‑permitting modeling of the current and future climate of North America. Climate Change, DOI 10.1007/s00382-016-3327-9

Page 14, Line 12: I do not think that it is 'sophisticated enough. . .'. Melting rates of particles aloft, for example, depend on the features of the particles themselves as well as temperature and moisture conditions.

Page 14, Line 15: Why did you not examine sounding information taken during freezing rain events? You could then more quantitatively assess how well the approach is handling particular instances. The lack of such validation is a major drawback in this article.

Page 14, around Line 25: Why not compare against previous studies on the climatological features?

Page 14, Line 30: Given the limitations of the dataset as you have mentioned, how

confident are you that you can 'reliably' address such questions?

Page 14, Line 31: Clarify what is meant by 'station scale analysis'.

Technical Corrections:

Page 2, Lines 5 and 7: The word 'where' is not correct in referring to an event 'in time'. This error was done in other places as well.

Page 2, Line 22: Another incorrect use of 'where'.

Page 13, Line 28: 'criteria'

––––––––––––––––––––––––––––––

---

## Author Comment (AC1) · 14 Oct 2016

**Referee #1, Dimitar Nikolov National Institute of Meteorology and Hydrology 1784 Sofia, Bulgaria**

*We would like to thank Dr. Nikolov for the in-depth review and detailed comments. His comments are in **bold** and our replies to the comments are in normal font. We included here and replied only the critical comments.*

…

**Specific comments**

**I would recommend the authors replace the references Rauber et al., 2000; Carrière et al., 2000 for the warm rain process (page 2) with (or add) the following two:**
- **Bocchieri, J., 1980: The objective use of upper air soundings to specify precipitation type. Mon. Wea. Rev., 108, 596–603.**
- **Huffman, G. J., and G. A. Norman Jr., 1988: The supercooled warm rain process and the specification of freezing precipitation. Mon. Wea. Rev., 116, 2172–2182.**

The references recommended by Dr. Nikolov are indeed the ones to first describe the phenomenon, and are better than Carrière et al. (2000) in this context. We suggest keeping Rauber et al. (2000) and replacing Carrière et al. (2000) with Bocchieri (1980), and Huffman and Norman (1988).

**I disagree with the decision of the authors to exclude most of the stations from eastern Europe. It seems that they have erroneously interpreted the explanations of Bezrukova et al. (2006) for the different definitions of FZRA events in these countries. Indeed, sometimes the icing due to supercooled clouds or fogs may deposit as glaze (wet growth process) and then the symbol for glaze is written down, but such a case will never be reported as freezing rain or freezing drizzle in the WMO weather codes. This ambiguity concerns mostly the local meteorological archives where additional control is needed to distinguish between both events. The weather codes 24, 56, 57, 66 and 67 are not affected at all. By this reason the authors (of Bezrukova et al., 2006) have decided to restrict only to the WMO codes.**

Indeed, it seems to be likely that we misinterpreted the methodology of Bezrukova et al. (2006). However, based on our results, it is clear that when all stations from the domain (i.e. including eastern stations) are used in calibration of the FMICLIM algorithm, we get a prominent overestimation of algorithm-based total number of FZRA events in eastern parts and underestimation in other parts of Europe for some reason. When the eastern stations were excluded from calibration, the validation results enhanced slightly (and the underestimation of the number of FZRA events disappeared elsewhere, because of the unbiasing nature of the calibration procedure).

We agree that using all relevant data should be preferred. On the other hand, prior to further studies, it would be very important to explain why we get so different results in eastern Europe compared to other areas. Because the method (FMICLIM) and data (ERA-Interim) are the same over the domain, the only imaginable reasons, we think, are (1) local conditions favouring/inhibiting the formation of FZRA and/or (2) differences in how FZRA is observed in different countries. Because the eastern area we excluded contains both flat terrain and mountains, (2) might be more likely than (1). Moreover, it is perhaps unlikely that several totally independent observational data sets of FZRA would exist: maybe at least partly same data (or same observers) were used in datasets presented by Bezrukova et al. (2006). However, we admit that our reasoning here is rather speculative. Additionally, the division to 'eastern' and 'other' stations is rather coarse and should be refined in future.

We suggest modifying the text (Page 4, lines 5 -- 20) so that the misinterpretation of the paper of Bezrukova et al. (2006) is corrected, but keeping the selected set of stations as it is.

**The authors of the manuscript have also filtered the data outside the interval – 30oC +10oC, which seems to be too wide. Most often FZRA occur in the interval – 10oC 0oC, so an appropriate interval for filtering, in my opinion, would be –15oC + 5oC. This would prevent to a certain extent from misclassification of ice pellets as FZRA or FZDR.**

This is true. We suggest (1) testing how much data is filtered out using the proposed stricter interval, and if a large proportion is filtered out, (2) testing how much validation scores are altered. If major enhancements in validation scores are found in (2), we should perhaps also consider rerunning the calibration procedure.

**The finding that the altitude does not contribute to the explain variance is somehow surprising for me. One would expect that the number of FZRA and their duration would decrease with the altitude because of the decreasing of the depth of the near-surface cold layer and FZRA aloft should be even more rare event**

**than the FZRA at the ground. However, mountain ranges mostly caused cold air damming which is difficult to be recognized in data sets with coarse resolution.**

Actually, altitude is, as well, correlated with the total number of observed FZRA events, but slightly less strongly than the distance to the coastline. Since the altitude and distance are themselves quite strongly correlated, including both variables to the model did not enhance the result much compared to using just one of them. And because the distance was slightly better in explaining the variance, we decided to use it.

We suggest adding this clarification to the text.

**The vertical resolution of the FMINWP seems to be not very appropriate for detailed representation of the vertical profiles of the relative humidity and the air temperature, which would affect the correct estimation of the near-surface cold layer and the melting layer above. It can be seen that an increasing of the resolution is foreseen as future work and this would be very helpful.**

Using a coarse vertical resolution, despite the drawbacks, was partly selected because output data from climate models, which we are going to analyse next, is commonly available in a rather coarse vertical resolution as well, and one purpose of this paper was to show that some kind of results can be achieved also by that way.

We suggest adding this clarification to the Introduction and Conclusions parts.

**The minimum acceptable cold layer depth has been significantly increased by the calibration procedure – from 130 meters up to 400 meters. This seems very reasonable because of the large size of the investigated area and variable weather conditions. For example Bernstein reported values of the near-surface cold layer in USA between 100 and 1400 meters, the minimums being between 100 and 300 meters.**
  ● **Bernstein, B., 2000: Regional and local influences on freezing drizzle, freezing rain, and ice pellet events. Wea. Forecasting, 15, 485–508.**

We thank Dr. Nikolov for this information. We suggest completing the text based on this comment, and adding the new provided reference.

**…**

**Very interesting results are presented in the paragraph 3.3 Climatology of freezing rain in Europe. However, the finding for a maximum in the annual number**

**of events over the Carpathian mountain sounds surprisingly for me. It would be useful if the altitude of these regions is given.**

This is an important comment, and reveals one weakness in the results: the FMICLIM algorithm cannot detect FZRA reliably at high altitudes (perhaps >2000m or maybe >1750m), because there the pressure levels used are too few to represent the cold layer -- melting layer structure. We suggest replotting the maps (Figs. 8 and 9) using a mask which hides the suspicious high altitude results. Also, analyses including the high altitude results should be recalculated (e.g. Fig. 7) by excluding the high-altitude data.

We also suggest trying to include elevation information to Fig. 1, but there is a risk that readability of the figure deteriorates. If it gets too low, we perhaps should not include elevation.

**Technical comments**

**I have encountered only two small misprints – on page 9, third row – the FMICLIM is written wrongly and on page 11, third row is written "The Carpathian...".**

We suggest correcting this.

**Conclusions**

**…**

**The size, quality and readability of each figure is adequate to the type and quantity of data presented, except for figure 5 which could be a little bit larger.**

We suggest enlarging the figure, or replotting it so that markings and details are clearer.

**The authors give proper credit to previous and related work with a small oversight of two references for the warm rain process.**

We suggest adjusting the references as proposed previously.

**…**

**The authors should only take into account that they could use for future investigation the stations in eastern Europe with no restrictions, as far as they utilize the international weather WMO codes.**

We agree that using all relevant data should be preferred, but we suggest keeping the restrictions as they are in the paper. We admit that the evidence to support that decision is not as solid as we would like it to be, but it is the 'least bad' choice in our opinion.

**…**

---

## Author Comment (AC2) · 14 Oct 2016

**Referee #2, anonymous**

*We would like to thank the Referee #2 for the in-depth review and detailed comments. His/Her comments are in **bold** and our replies to the comments are in normal font. We included here and replied only the critical comments.*

…

**Specific Comments**

**Page 1, Line 17: There are many instances of short but 'heavy' freezing rain.**

We suggest removing 'short-lived'.

**Page 2, Line 12: A recent climatology over parts of northern Eurasia has been completed:**
  - **Groisman, P.Ya., O. N. Bulygina, X. Yin, R. S. Vose, S. K. Gulev, I. Hanssen-Bauer, and E. Førland 2016: Recent changes in the frequency of freezing precipitation in North America and Northern Eurasia. Environ. Res. Lett. 11, 045007.**

We suggest familiarizing ourselves with this work and, if found relevant, including it as a reference.

**Page 3, Line 27: 3-hourly reports are probably insufficient. Most freezing rain events occur at shorter time scales. Was any attempt made to at least 'estimate' how many events were uncounted by using hourly information as well?**

No such attempt was made since we did not have hourly observations available. Without good observations, too many assumptions are required for the estimation of the distribution of the short-term events. This would be an interesting exercise to do, but it is out of the scope of the present article, and indeed requires 1-hourly (or maybe even denser) observations.

**Page 3, Line 27: I may have missed this but how did you treat combinations of precipitation types? It is common for freezing rain to occur with ice pellets for example.**

According to WMO standards, only the weather with the largest code number is reported in the SYNOP present weather code. Therefore, if e.g. ice pellets (79) occur together with freezing rain (66, 67), only ice pellets are reported, and no information of the simultaneous FZRA is recorded at all. We calibrated the FMICLIM method to identify reported FZRA, and for that reason selected SYNOP codes 24 (freezing rain within past hour but not at observation time), 66 (light freezing rain) and 67 (moderate to heavy freezing rain) as target classes.

**Page 4, Line 3: A threshold of 80% is quite low. Why didn't you show the fraction of missing data during the cold season?**

We suggest showing the fraction of missing data during the cold season to help readers to estimate the number of missing FZRA observations in the total numbers of FZRA events.

**Page 4, Line 15: What fraction of observations was beyond these thresholds? Were any of the high valued temperatures associated with very low relative humidities that would lead to much lower wet bulb temperatures?**

We did not understand if the Referee #2 is asking what fraction of FZRA observations, or what fraction of all observations of surface air temperature is beyond the thresholds. Here, we assume that he/she means FZRA observations. In that case we suggest (1) testing how much FZRA data is filtered out by those thresholds, and (2) testing how much FZRA data is filtered out using the stricter interval proposed by Referee #1 (Dr. Nikolov). If a large proportion is filtered out in (2), we suggest (3) testing how much validation scores are altered. If major enhancements in validation scores are found in (3), we should also consider rerunning the calibration procedure.

**Page 5, Line 7: This scale is very large for freezing rain. It is quite common for these regions to be less.**

We are not quite sure if we  understood this comment correctly.

If the Referee #2 is referring to the coarse spatial resolution of the ERA-Interim reanalysis data, we agree that we would prefer to use spatially denser reanalyses. As said in the text (Page 13, Line 28), this is considered to be one part of the future work.

**Page 6, Line 3: Ice-initiated precipitation is not initially generated in the inversion aloft. What is the implication of assuming it is?**

The implication is that the algorithm does not identify those FZRA events. Those non-identified events are compensated in the calibration by adjustments of the temperature thresholds in different layers. These adjustments then lead to generation of erroneous FZRA events to some other time steps (i.e. false alarms), and to deterioration of the validation results.

We suggest adding a sentence about that in the revised paper.

**Page 11, Line 31: This paragraph is poorly worded and hard to follow.**

We suggest to rewrite the paragraph, for example as follows:

"In validation, the gridded meteorological dataset  is compared with the point-like surface observations. Each grid cell represents spatial means in the 0.7° resolution, while weather stations represent more local variability of the atmosphere. It is possible that in some cases FZRA has not been observed at a station although it has occurred rather nearby. Although hypothetical, this suggests that our estimates, derived from ERA-Interim, might at least occasionally represent the occurrence of FZRA inside the 0.7 grid cells better than the stations do."

We hope this suggestion is better worded and easier to follow.

**Page 12, Line 5: There are standard observing practices to identify freezing rain. Why is this so hard to do?**

As said in the text, simply "because the phenomenon can be easily confused with ordinary, non-freezing rain": For an observer, freezing rain looks the same by eye as non-freezing rain, and accumulation of ice might not be visible (1) in short-term events or (2) when snow covers objects on the ground. Besides, the observed 2m temperature, which is most commonly used by the observers to distinguish FZRA from non-freezing rain at stations, might not represent the temperature of the thick (maybe several hundreds meters) near-surface cold layer well enough in all cases, and knowing if the rain actually is supercooled or not is not so straightforward.

**Page 12, Line 8: You are associating 'minor' with short duration. On what basis? There can be severe impacts with durations smaller than 6 h and precipitation rates can be high as well..**

We suggest replacing that sentence with these:

"Additionally, short-term events, which are more common than longer ones (Ressler et al., 2012; Cortinas, 2000), might not be recorded in the 6-hourly observations. Short-term events are difficult to predict using spatially and temporally smoothed 6-hourly reanalysis data."

We also suggest correcting references to 'minor FZRA' in other parts of the text.

**Page 12, Line 11: Are the errors 'random'?**

They are 'random' in the sense that we try to predict station level variability using grid cell -level information. Also they are 'random' in the sense that we assume the observational errors to be random: equal amounts of (1) false identifications of FZRA events and (2) false rejections of FZRA events happen.

We suggest completing the sentence:

"To maximize the reliability of the observations, a large number of SYNOP stations was used, which is believed to average out random errors in calculation of spatially and/or temporally aggregated results, such as mean annual numbers of events in subgroups (Fig. 3)."

**Page 12 Line 25: Given the enormous smoothing at 70 km, maybe the authors should only consider analyses over 'flat regions'?**

This is a good comment. In addition, there is actually even a more important aspect to justify exclusion of the highest elevations from the analysis: the FMICLIM algorithm can not detect FZRA reliably at high altitudes (perhaps >2000m or maybe >1750m), because there the pressure levels used (925, 850, 700 hPa and 2-meter levels) are too few to represent the cold layer -- melting layer structure. We suggest replotting the maps (Figs. 8 and 9) using a mask which hides the suspicious high altitude results. Also, analyses including the high altitude results should be recalculated (e.g. Fig. 7) by excluding the high-altitude data.

**Page 12, Line 33: "Occasional misclassification"? How often did this occur?**

It happens sometimes, as shown in the Section 3.2.1 (SYNOP weather code classification) and in Fig. 5.

**Page 13, Line 3: To me, this section is too long and wordy. This is a long shopping list. What are the most important and feasible next steps? From my perspective, some of these should be done within this article.**

We agree to order the list based on the expected importance of the future steps. However, we prefer to present all the suggested steps, because we think that they are all necessary to get more accurate results.

**As well, a recent article (Liu et al., 2016) pointed out that precipitation at the surface (including freezing rain) is calculated directly from the model's microphysical package without needing the approach used here. Isn't that the best way forward?**
  - **Liu et al., 2016: Continental scale convection permitting modeling of the current and future climate of North America. Climate Change, DOI 10.1007/s00382-016-3327-9**

This is of course the optimal solution. However, the precipitation type is not included in the output variables of ERA-Interim.

**Page 14, Line 12: I do not think that it is 'sophisticated enough...'. Melting rates of particles aloft, for example, depend on the features of the particles themselves as well as temperature and moisture conditions.**

We suggest removing 'sophisticated enough'.

**Page 14, Line 15: Why did you not examine sounding information taken during freezing rain events? You could then more quantitatively assess how well the approach is handling particular instances. The lack of such validation is a major drawback in this article.**

We agree that the division of uncertainty to method-dependent and ERA-Interim-dependent components would be extremely informative. Examining sounding information is suggested to be given the highest priority in the list of the future steps (Sec. 4.2).

However, the 'perfect data' approach would only give a sort of upper limit estimation of the performance of the algorithm, as, with our gridded data, the algorithm has to operate with the limited number of pressure levels for example. We were interested in the overall capability of the method *and* the ERA-Interim reanalysis in describing the FZRA climatology, and found that together they give a rather good estimate, when aggregated results are evaluated (e.g. Fig. 3). We also got and presented information about the total uncertainty consisting of both uncertainty components.

**Page 14, around Line 25: Why not compare against previous studies on the climatological features?**

We suggest adding some sentences to this part of the text, even though climatological studies of FZRA in Europe are almost inexistent.

**Page 14, Line 30: Given the limitations of the dataset as you have mentioned, how confident are you that you can 'reliably' address such questions?**

In that sentence, we (implicitly) refer to Fig. 3, and there we can be quite confident because our results are backed by the observations. There are other results that we are less confident and which are a motivation for further development of the FZRA detection methodology.

**Page 14, Line 31: Clarify what is meant by 'station scale analysis'.**

We suggest replacing 'station scale' with 'station level' in all occurrences (i.e. in the Conclusions and the Abstract sections).

Station level analysis means comparisons of raw modelled and observed time series of FZRA in all individual stations. No aggregation of data temporally or spatially is applied prior to analysis.

**Technical Corrections**

**Page 2, Lines 5 and 7: The word 'where' is not correct in referring to an event 'in time'. This error was done in other places as well.**

We suggest correcting this.

**Page 2, Line 22: Another incorrect use of 'where'.**

We suggest correcting this.

**Page 13, Line 28: 'criteria'**

We suggest correcting this.

---

## Author Response (AR1)

**Author's Response**

*to the comments of Referees and the Editor of NHESS regarding the discussion paper*

Kämäräinen, M., Hyvärinen, O., Jylhä, K., Vajda, A., Neiglick, S., Nuottokari, J., and Gregow, H.: A method to estimate freezing rain climatology from ERA-Interim reanalysis over Europe, Nat. Hazards Earth Syst. Sci. Discuss., doi:10.5194/nhess-2016-225, in review, 2016.

**In Helsinki, 4 January 2016**
**Matti Kämäräinen**

**Table of Contents**

**1. A point-by-point response to the reviews**

Please note that the comments given here by the author are partly different than the comments that were sent to the 'Public discussion' of the paper.

**Referee #1, Dimitar Nikolov**

**National Institute of Meteorology and Hydrology 1784 Sofia, Bulgaria**

*We would like to thank Dr. Nikolov for the in-depth review and detailed comments. His comments are in **bold** and our replies to the comments are in normal font. We included here and replied only the critical comments.*

…

**Specific comments**

**I would recommend the authors replace the references Rauber et al., 2000; Carrière et al., 2000 for the warm rain process (page 2) with (or add) the following two:**

- **Bocchieri, J., 1980: The objective use of upper air soundings to specify precipitation type. Mon. Wea. Rev., 108, 596–603.**
- **Huffman, G. J., and G. A. Norman Jr., 1988: The supercooled warm rain process and the specification of freezing precipitation. Mon. Wea. Rev., 116, 2172–2182.**

We kept Rauber et al. (2000) and replaced Carrière et al. (2000) with Bocchieri (1980), and Huffman and Norman (1988).

**I disagree with the decision of the authors to exclude most of the stations from eastern Europe. It seems that they have erroneously interpreted the explanations of Bezrukova et al. (2006) for the different definitions of FZRA events in these countries. Indeed, sometimes the icing due to supercooled clouds or fogs may deposit as glaze (wet growth process) and then the symbol for glaze is written**

**down, but such a case will never be reported as freezing rain or freezing drizzle in the WMO weather codes. This ambiguity concerns mostly the local meteorological archives where additional control is needed to distinguish between both events. The weather codes 24, 56, 57, 66 and 67 are not affected at all. By this reason the authors (of Bezrukova et al., 2006) have decided to restrict only to the WMO codes.**

Indeed, it seems to be likely that we misinterpreted the methodology of Bezrukova et al. (2006). In December 2016 we contacted Dr. Bezrukova via email and she confirmed that in Russia, the SYNOP station observations are performed following the WMO standards.

Based on our results, though, it is clear that when all stations from the domain (i.e. including eastern stations) are used in calibration of the FMICLIM algorithm, we get a prominent overestimation of algorithm-based total number of FZRA events in eastern parts and underestimation in other parts of Europe for some reason.

Because we could not identify whether the reason to the differences in predicted and observed values originates from the reanalysis or from observations, and following the suggestions of Drs. Nikolov and Bezrukova, we extended the station set to include also the eastern stations. We recalibrated the algorithm with all relevant stations and modified the text, tables, and figures accordingly. The most important effect was found in the mean values, so that recalibrated algorithm produces less freezing rain than the previous version. Additionally the validation results deteriorated: some metrics, e.g. the correlation between the spatially averaged observed and predicted annual mean numbers is lower after calibration than before it.

**The authors of the manuscript have also filtered the data outside the interval –30oC +10oC, which seems to be too wide. Most often FZRA occur in the interval –10oC 0oC, so an appropriate interval for filtering, in my opinion, would be –15oC + 5oC. This would prevent to a certain extent from misclassification of ice pellets as FZRA or FZDR.**

We recalibrated the algorithm using this new stricter filter, which enhanced the validation results slightly. The enhancement was not as significant as the deterioration of the results following the inclusion of the eastern stations.

**The finding that the altitude does not contribute to the explain variance is somehow surprising for me. One would expect that the number of FZRA and their**

**duration would decrease with the altitude because of the decreasing of the depth of the near-surface cold layer and FZRA aloft should be even more rare event than the FZRA at the ground. However, mountain ranges mostly caused cold air damming which is difficult to be recognized in data sets with coarse resolution.**

The text was clarified so that the correlation of distance and elevation is more clear. The highest elevations, where freezing rain is very rare, were masked out from the maps and analyses, because the algorithm can not work there due to lack of pressure levels.

**The vertical resolution of the FMINWP seems to be not very appropriate for detailed representation of the vertical profiles of the relative humidity and the air temperature, which would affect the correct estimation of the near-surface cold layer and the melting layer above. It can be seen that an increasing of the resolution is foreseen as future work and this would be very helpful.**

Using a coarse vertical resolution, despite the drawbacks, was partly selected because output data from climate models, which we are going to analyse next, is commonly available in a rather coarse vertical resolution as well, and one purpose of this paper was to show that some kind of results can be achieved also by that way.

Some clarifications were added to the Introduction and Conclusions parts.

**The minimum acceptable cold layer depth has been significantly increased by the calibration procedure – from 130 meters up to 400 meters. This seems very reasonable because of the large size of the investigated area and variable weather conditions. For example Bernstein reported values of the near-surface cold layer in USA between 100 and 1400 meters, the minimums being between 100 and 300 meters.**
  ● **Bernstein, B., 2000: Regional and local influences on freezing drizzle, freezing rain, and ice pellet events. Wea. Forecasting, 15, 485–508.**

We completed the text based on this comment, and added the new provided reference.

**…**

**Very interesting results are presented in the paragraph 3.3 Climatology of freezing rain in Europe. However, the finding for a maximum in the annual number of events over the Carpathian mountain sounds surprisingly for me. It would be useful if the altitude of these regions is given.**

The highest elevations, where freezing rain is very rare, were masked out from the maps, because the algorithm can not work there due to lack of pressure levels. Also, elevation information was included in the Fig. 1.

**Technical comments**

**I have encountered only two small misprints – on page 9, third row – the FMICLIM is written wrongly and on page 11, third row is written "The Carpathian...".**

This was corrected.

**Conclusions**

**…**

**The size, quality and readability of each figure is adequate to the type and quantity of data presented, except for figure 5 which could be a little bit larger.**

The figure was enlarged slightly.

**The authors give proper credit to previous and related work with a small oversight of two references for the warm rain process.**

The references were adjusted based on this comment.

**…**

**The authors should only take into account that they could use for future investigation the stations in eastern Europe with no restrictions, as far as they utilize the international weather WMO codes.**

The method was recalibrated, the results adjusted, and figures replotted using all relevant stations.

**…**

**Referee #2, anonymous**

*We would like to thank the Referee #2 for the in-depth review and detailed comments. His/Her comments are in **bold** and our replies to the comments are in normal font. We included here and replied only the critical comments.*

…

**Specific Comments**

**Page 1, Line 17: There are many instances of short but 'heavy' freezing rain.**

'Short-lived' was removed from the text.

**Page 2, Line 12: A recent climatology over parts of northern Eurasia has been completed:**
  ● **Groisman, P.Ya., O. N. Bulygina, X. Yin, R. S. Vose, S. K. Gulev, I. Hanssen-Bauer, and E. Førland 2016: Recent changes in the frequency of freezing precipitation in North America and Northern Eurasia. Environ. Res. Lett. 11, 045007.**

This work was included as a reference.

**Page 3, Line 27: 3-hourly reports are probably insufficient. Most freezing rain events occur at shorter time scales. Was any attempt made to at least 'estimate' how many events were uncounted by using hourly information as well?**

No such attempt was made since we did not have hourly observations available. Without good observations, too many assumptions are required for the estimation of the distribution of the short-term events. This would be an interesting exercise to do, but it is out of the scope of the present article, and indeed requires 1-hourly (or maybe even denser) observations.

**Page 3, Line 27: I may have missed this but how did you treat combinations of precipitation types? It is common for freezing rain to occur with ice pellets for example.**

According to WMO standards, only the weather with the largest code number is reported in the SYNOP present weather code. Therefore, if e.g. ice pellets (79) occur together with freezing rain (66, 67), only ice pellets are reported, and no information of the simultaneous FZRA is recorded at all. We calibrated the FMICLIM method to identify reported FZRA, and for that reason selected SYNOP codes 24 (freezing rain within past hour but not at observation time), 66 (light freezing rain) and 67 (moderate to heavy freezing rain) as target classes.

**Page 4, Line 3: A threshold of 80% is quite low. Why didn't you show the fraction of missing data during the cold season?**

The fraction of missing data during the cold season is now presented in the text to help readers to estimate the number of missing FZRA observations in the total numbers of FZRA events.

**Page 4, Line 15: What fraction of observations was beyond these thresholds? Were any of the high valued temperatures associated with very low relative humidities that would lead to much lower wet bulb temperatures?**

The algorithm was recalibrated using the stricter temperature intervals, which enhanced the calibration results slightly.

The combination of high temperatures and very low relative humidity is quite rare in the observational data:

- 21% of the freezing rain happened with above-zero temperatures
- 5% happened with T2m > 0C AND RH2m < 90%
- 1% happened with T2m > 0C AND RH2m < 80%
- 0.1% happened with T2m > 0C AND RH2m < 60%

Of course, using wet bulb temperatures instead of air temperatures would most probably enhance the results. This was added to the 'Future work' section in the text.

**Page 5, Line 7: This scale is very large for freezing rain. It is quite common for these regions to be less.**

We would prefer to use spatially and temporally denser reanalyses. As said in the text, this is considered to be one part of the future work.

**Page 6, Line 3: Ice-initiated precipitation is not initially generated in the inversion aloft. What is the implication of assuming it is?**

The implication is that the algorithm does not identify those FZRA events. Those non-identified events are compensated in the calibration by adjustments of the temperature thresholds in different layers. These adjustments then lead to generation of erroneous FZRA events to some other time steps (i.e. false alarms), and to deterioration of the validation results.

The separation of moist (precipitation generation) layer and melting layer are mentioned in the 'Future work' section.

**Page 11, Line 31: This paragraph is poorly worded and hard to follow.**

The paragraph was restructured.

**Page 12, Line 5: There are standard observing practices to identify freezing rain. Why is this so hard to do?**

As said in the text, simply "because the phenomenon can be easily confused with ordinary, non-freezing rain": For an observer, freezing rain looks the same by eye as non-freezing rain, and accumulation of ice might not be visible (1) in short-term events or (2) when snow covers objects on the ground. Besides, the observed 2m temperature, which is most commonly used by the observers to distinguish FZRA from non-freezing rain at stations, might not represent the temperature of the thick (maybe several hundreds meters) near-surface cold layer well enough in all cases, and knowing if the rain actually is supercooled or not is not so straightforward.

**Page 12, Line 8: You are associating 'minor' with short duration. On what basis? There can be severe impacts with durations smaller than 6 h and precipitation rates can be high as well..**

References to 'minor FZRA' in the text were removed.

**Page 12, Line 11: Are the errors 'random'?**

They are 'random' in the sense that we try to predict station level variability using grid cell level information. Also they are 'random' in the sense that we assume the observational errors to be random so that equal amounts of (1) false identifications of FZRA events and (2) false rejections of FZRA events happen.

The paragraph was completed to be more clear.

**Page 12 Line 25: Given the enormous smoothing at 70 km, maybe the authors should only consider analyses over 'flat regions'?**

The highest elevations were excluded from the analysis and masked out from the maps.

**Page 12, Line 33: "Occasional misclassification"? How often did this occur?**

It happens sometimes, as shown in the Section 3.2.1 (SYNOP weather code classification) and in Fig. 6.

**Page 13, Line 3: To me, this section is too long and wordy. This is a long shopping list. What are the most important and feasible next steps? From my perspective, some of these should be done within this article.**

The list was ordered and modified based on the expected importance of the future steps.

**As well, a recent article (Liu et al., 2016) pointed out that precipitation at the surface (including freezing rain) is calculated directly from the model's microphysical package without needing the approach used here. Isn't that the best way forward?**
  ● **Liu et al., 2016: Continental scale convection permitting modeling of the current and future climate of North America. Climate Change, DOI 10.1007/s00382-016-3327-9**

This is of course the optimal solution. However, the precipitation type is not included in the output variables of ERA-Interim (or in the publicly available climate model output datasets which we are going to analyse next).

**Page 14, Line 12: I do not think that it is 'sophisticated enough...'. Melting rates of particles aloft, for example, depend on the features of the particles themselves as well as temperature and moisture conditions.**

'Sophisticated enough' was removed.

**Page 14, Line 15: Why did you not examine sounding information taken during freezing rain events? You could then more quantitatively assess how well the approach is handling particular instances. The lack of such validation is a major drawback in this article.**

Examining sounding information was given a high priority in the list of the future steps (Sec. 4.2).

**Page 14, around Line 25: Why not compare against previous studies on the climatological features?**

The most relevant studies in this context, i.e. Bezrukova et al., 2006, Carriere et al., 2000, and Groisman et al., 2016, can not be compared to our results because of different methodology (Bezrukova et al., 2006 and Groisman et al., 2016) or too a short period used (Carriere et al., 2000).

**Page 14, Line 30: Given the limitations of the dataset as you have mentioned, how confident are you that you can 'reliably' address such questions?**

In that sentence, we (implicitly) refer to Fig. 3, and there we can be quite confident because our results are backed by the observations. There are other results that we are less confident and which are a motivation for further development of the FZRA detection methodology.

**Page 14, Line 31: Clarify what is meant by 'station scale analysis'.**

'Station scale' was replaced with 'station level' in all occurrences (i.e. in the Conclusions and the Abstract sections).

Station level analysis means comparisons of raw modelled and observed time series of FZRA in all individual stations. No aggregation of data temporally or spatially is applied prior to analysis.

**Technical Corrections**

**Page 2, Lines 5 and 7: The word 'where' is not correct in referring to an event 'in time'. This error was done in other places as well.**

This was corrected.

**Page 2, Line 22: Another incorrect use of 'where'.**

This was corrected.

**Page 13, Line 28: 'criteria'**

This was corrected.

**2. A list of all relevant changes made in the manuscript**

1. The calibration was recalculated using all relevant weather stations, i.e. the eastern stations which were previously excluded, we included in the calibration.
2. A new figure was added to clarify the east-west bias results in the station network.
3. Figures were replotted using the newly calibrated algorithm, and the highest elevations were masked out in the maps.
4. Numbers in the text and tables were corrected after recalibration.
5. The chapter 'Number of freezing rain events at weather stations' was partly restructured and the detailed explanations of changes in statistics were removed, because the calibration did not change them anymore after the adjustments in calibration station set.
6. The chapter 'Future work' was partly restructured.

**3. A marked-up manuscript version**

Please note that the figures in this marked-up version are partly old, and partly wrongly numbered. Additionally the 'latexdiff' software adds some extra lines in the beginning of the document for some reason.

The  highest elevations were excluded (gray color) because of  larger uncertainties in  FZRA detection. figure.9

[revised manuscript text omitted]
  to the coastline and elevation were the best ones in explaining the variance ( e.g. the adjusted $R^2$ for distance was 0.06, $p \ll$

5  .001).  Because elevation and distance are rather strongly correlated, using both or their interaction term did not improve the results substantially,  that is, it did not change  the adjusted $R^2$. For this reason, only the distance to the nearest coastline was  used for the classification. The stations were  grouped to "coastal" (0–140 km), "semi-coastal" (140–330 km), and "continental" (>330 km). Boundaries

10 of classes , shown in Fig. 1, were selected so that each group contained equal number  of validation stations.

**2.2   ERA-Interim reanalysis data**

Relative humidity and temperature from 925, 850, 700 hPa and 2-meter levels, surface pressure, and precipitation of the ERA-Interim reanalysis dataset (Dee et al., 2011) were used as a predictor data.

15  The variables were derived from the 6-hourly analysis part of ERA-Interim except that for precipitation the 6-hourly forecasted part was used and the 12-hourly precipitation sums in were transformed into 6-hourly sums. 
[revised manuscript text omitted]

25  the stations where the large values occur. In the continental group (Fig. 5d), the curve is  _horizontal, or even negatively correlated,_ for all values. In the coastal group the RMS error is the smallest but there is an  _underestimation (ME < 0)_, while for the continental group the RMS error is the largest and there is an  _overestimation (ME > 0)_. Smaller RMS error in coastal areas can be partly explained by the lack of large values that would contribute to RMS.

30   _The spatial averages were computed over all the 293 stations in the calibration set and separately for the stations in the coastal, semi-coastal, and continental subgroups. Because of the uncertainties in the SYNOP weather station observations or in the ERA-Interim reanalysis, calibration had only minor effect on the spatially averaged annual numbers of FZRA events inside the subgroups or over the whole station network_ (Fig. ).

shown) 3 and Table 3). Compared to observations, the calibrated and uncalibrated algorithms both underestimate slightly the coastal mean number and overestimate the continental mean number of events, the semi-coastal group being better modelled. Both versions of the algorithm overestimate the standard deviation in all groups and at individual stations. The annual numbers of FZRA events are reproduced better as spatial averages across the weather stations than at individual sites. The RMS error of spatial averages is smaller than the standard deviation of observations, except for the continental group, implying results in general are better using FMI$_{CLIM}$ or FMI$_{NWP}$ than just using the climatology. For individual sites, however, the RMS error is slightly worse than the standard deviation of observations.

An area of underestimation was found in southern Germanyand in Austriacan be seen e.g. in Germany, Czech Republic, and in most stations of Poland, while overestimation mostly happens in the northern and eastern validation stations . As explained above, two reasons why stations were excluded from the validation were the low observed number of FZRA events andthe different definition of observed freezing rain. Most of the stations that were excluded from the validation due to the low observed number of FZRA events are located on coasts. This (Figs. 1 and 4). These biases are spatially homogeneous, independent of algorithm versions (not shown), and do not change in time. The lack of freezing rain, mostly seen in coastal stations, 
[revised manuscript text omitted]

– If the bias structure between the eastern and central European SYNOP stations (Fig. 4) is caused by observational uncertainties, and not by reanalysis or algorithm-dependent uncertainties, identification and rejection of low-quality stations could enhance the calibration and validation processes. Slightly better validation scores between observations and prediction was achieved when eastern stations were excluded from the calibration (not shown).

– Increasing the vertical resolution of the $FMI_{NWP}$ algorithm would be helpful, as small differences in vertical layers easily affect the result (Stewart et al., 2015).

– Validation of the vertical temperature profile of the reanalysis data against observational soundings would allow the division of the uncertainty into method-dependent and data-dependent components.

[revised manuscript text omitted]

*Acknowledgements.* This work was partly funded by the European Union's Seventh Programme for research, technological development and demonstration under the RAIN project (Risk Analysis of Infrastructure Networks in response to extreme weather; http://rain-project.eu/; grant agreement N° 608166). The work has also received funding from the  State Nuclear Waste Management Fund in Finland and from the Swedish Radiation Safety Authority through the EXWE project (Extreme weather and nuclear power plants) of the SAFIR2018 program (The Finnish Nuclear Power Plant Safety Research Programme 2015-2018; http://safir2018.vtt.fi). We thank Sami Niemelä for helping with MARS data retrieval, Tiina Ervasti and Curtis Wood for improving the grammar of the article, and Juulia Lahdenperä for commenting the text. Some of the results were presented in annual meeting of the European Meteorological Society (EMS) in September 2015 by Kämäräinen et al. (2015).

[revised manuscript text omitted]

**Figure 7.** Vertical profiles of temperature (°C, left column) and relative humidity (%, right column) of ERA-Interim at weather station locations. 5%–95% range (cyan), 25%–75% range (blue) and mean (red) are shown. Top row: profiles when FZRA was reported in SYNOP messages ( 11000 events in total). Middle row: FZRA profiles according to the calibrated FMI$_{CLIM}$ algorithm ( 11000 events). Bottom row: profiles where both the FMI$_{CLIM}$ algorithm and observations indicated FZRA ( 2300 events).

[Figure]

**Figure 8.** Probability of duration (left column) and spatial extent (right column) of FZRA events at station locations (top row) and in all grid cells (bottom row) according to the detection algorithms (blue, red) and observations (black).

[Figure]

The  highest elevations were excluded (gray color) because of  larger uncertainties in  FMI$_{CLIM}$  FZRA  detection.

**Figure 9.** (a) Mean annual number of FZRA events and (b) maximum duration of events in the 1979-2014 study period. FMI$_{CLIM}$ algorithm is applied to the ERA-Interim reanalysis data.

The  highest elevations were excluded (gray color) because of  larger uncertainties in  FMI$_{CLIM}$  FZRA  detection.

[Figure]

**Figure 10.** The monthly climatology of FZRA in 1979–2014 according to the FMI$_{CLIM}$ algorithm, applied to the ERA-Interim. The average annual number of 6-hourly FZRA cases is shown. The highest elevations were excluded (gray color) because of larger uncertainties in FZRA detection.

**Table 1.** Uncalibrated (upper row) and calibrated (bottom), optimal values of threshold parameters in the 29-year calibration periods. Mean values of the optimal values are shown, computed for calibration periods using the sample variance of period values. Mean values are used in the final analysis of the gridded dataset. $h^{thr}_{cold}$ = minimum cold layer depth; $RH^{thr}_{melt}$ and $T^{thr}_{melt}$ = minimum humidity and minimum temperature in the melting layer; $T^{thr}_{cold}$ = maximum cold layer temperature; and $Pr^{thr}$ = minimum surface precipitation rate.

| | $h^{thr}_{cold}$ (hPa) > | $RH^{thr}_{melt}$ (%) > | $T^{thr}_{melt}$ (°C) > | $T^{thr}_{cold}$ (°C) < | $Pr^{thr}$ (mm 6h$^{-1}$) > |
|---|---|---|---|---|---|
| FMI$_{NWP}$ | 15 | 90 | 0 | 0 |  0.05 |
| FMI$_{CLIM}$ |  69 |  89 |  -0.64 |  0.09 |  0.39 |

**Table 2.** The cross-validation measures and scores in 7-year validation periods when predicted 6-hourly FZRA result is compared against observed 6-hourly events. Mean values and standard errors, computed for validation periods using the sample variance of period values, are shown. See text and Appendix A for definitions of measures and scores.

| | FMI$_{NWP}$ | FMI$_{CLIM}$ | FMI$_{CLIM}$ - FMI$_{NWP}$ |
|---|---|---|---|
| CSI |  0.109 ± 0.005 |  0.118 ± 0.004 |  0.009 ± 0.001 |
| SEDI |  0.65 ± 0.01 |  0.66 ± 0.01 |  0.014 ± 0.002 |
| HSS |  0.196 ± 0.009 |  0.211 ± 0.007 |  0.014 ± 0.002 |
| $a$ | 290 ± 20 430 ± 40 | 340 ± 20 460 ± 40 |  30 ± 7 |
| $b$ |  1730 ± 150 |  1690 ± 160 |  -40 ± 60 |
| $c$ |  1720 ± 100 |  1690 ± 100 |  -30 ± 7 |
| H |  0.197 ± 0.013 |  0.212 ± 0.015 |  0.015 ± 0.003 |
| F |  0.00063 ± 0.00005 |  0.00062 ± 0.00006 |  -0.00011 ± 0.00002 |
| B | 1.00 ± 0.05 |  1.00 ± 0.07 |  0.00 ± 0.03 |

**Table 3.** Statistics calculated from numbers presented in Fig. 3. Correlation coefficient of algorithm results compared to observations ($r$), mean value ($\overline{x}$), standard deviation ($s$), and RMS error (RMS) of annual mean numbers of FZRA cases per station averaged over all stations, averaged over groups based on distance to the nearest coastline, and in individual stations are shown.

| | FMI$_{NWP}$ | | | | FMI$_{CLIM}$ | | | | Observa |
|---|---|---|---|---|---|---|---|---|---|
| | $r$ | $\overline{x}$ | $s$ | RMS | $r$ | $\overline{x}$ | $s$ | RMS | $\overline{x}$ |
| All stations | 0.90 |  1.03 |  0.52 |  0.25 |  0.88 |  1.03 |  0.51 |  0.26 |  1.03 |
| Coastal |  0.84 |  0.71 |  0.39 |  0.23 |  0.86 |  0.72 |  0.37 |  0.20 |  0.80 |
| Semi-coastal |  0.90 |  1.01 |  0.53 |  0.24 |  0.86 |  1.01 |  0.54 |  0.28 |  1.02 |
| Continental |  0.83 |  1.26 |  0.69 |  0.41 |  0.83 |  1.26 |  0.72 |  0.44 |  1.20 |
| Individual stations |  0.38 |  1.03 |  1.55 |  1.65 |  0.38 |  1.03 |  1.58 |  1.66 |  1.03 |

**Table A1.** Contingency table of the comparison between observations and the algorithm. The symbols $a$-$d$ represent the different number of FZRA events observed to occur in each category.

| | Observation | |
|---|---|---|
| Algorithm | Freezing rain | No freezing rain |
| Freezing rain | $a$ (Hit) | $b$ (False Alarm) |
| No freezing rain | $c$ (Miss) | $d$ (Correct Rejection) |